# Non-unitary quantum many-body dynamics using the Faber polynomial method

Rafael D. Soares[1,2*] and Marco Schirò[2]

**1** Université Paris-Saclay, CNRS, LPTMS, 91405 Orsay, France
**2** JEIP, UAR 3573 CNRS, Collège de France, PSL Research University,
11 Place Marcelin Berthelot, 75321 Paris Cedex 05, France

⋆ rafael.diogo-soares@universite-paris-saclay.fr

## Abstract

Efficient numerical methods are still lacking to probe the unconventional dynamics of quantum many-body systems under non-unitary evolution. In this work, we use Faber polynomials to numerically simulate both the dynamics of non-Hermitian systems and the quantum jumps unravelling of the Lindblad dynamics. We apply the method to the non-interacting and interacting Hatano-Nelson models evolving from two different setups: i) a Néel state, and ii) a domain wall. In the first case, we study how interactions preserve the initial magnetic order against the skin effect. In the second example, we present numerical evidence of the existence of an effective hydrodynamic description for the domain-wall melting problem in the non-interacting limit. Additionally, we investigate both the conditional and unconditional dynamics of the quantum jump unravelling in two quantum spin chains, which exhibit either the non-Hermitian or the Liouvillian skin effect. This numerical method inherently generalises the well-established method based on Chebyshev polynomials to accommodate non-Hermitian scenarios.

# 1 Introduction

In recent years, the scientific community has shown a growing interest in elucidating the distinctive characteristics of many-body quantum systems subjected to effective non-unitary dynamics. In quantum mechanics, non-unitary dynamics typically arises when a closed quantum system interacts with an external environment, leading to dissipation, decoherence, or wave function collapse that disrupts the usual unitary Schrödinger evolution. While a fully microscopic description of both environment and system is a daunting task, in many cases of experimental and theoretical interest, one can assume the dynamics of the environment to be sufficiently fast, allowing for the derivation of a local-in-time, Markovian, non-unitary evolution for the system of interest [1, 2].

Different types of Markovian open quantum system dynamics have been considered in the literature. A first relevant example is provided by systems that evolve according to the Lindblad master equation [3–5]. Here, the evolution of the system density matrix is generated by a non-unitary (super)operator, the Lindbladian. Many-body versions of Lindblad master equations have been studied in a number of contexts and with different objectives, from dissipative phase transitions [6,7] to quantum transport [8]. Another class of non-unitary dynamics arises for continuously monitored quantum systems [2,9,10], whose stochastic evolution - a so-called quantum trajectory [11] - is described by a non-unitary unravelling of the Lindblad master equation [12]. In particular, under the quantum jump unravelling, a deterministic non-unitary evolution is driven by a non-Hermitian Hamiltonian between stochastic measurements [13–15]. Recent works have raised interest in possible phase transitions in the entanglement structure of those quantum many-body trajectories [16–22].

Finally, as a last example of non-unitary Markovian dynamics, we can consider the one generated by a purely non-Hermitian Hamiltonian. Non-Hermitian physics emerges intrinsically in various domains of physics beyond quantum physics, encompassing photonics [23], hydrodynamics [24] and active matter [25, 26]. In the context of open quantum systems, a non-Hermitian evolution can be obtained by post-selecting the quantum trajectories corresponding to no quantum jumps, i.e. the no-click limit [27]. Non-Hermitian quantum systems show anomalous static and dynamical properties, which are attracting widespread interest. Among these, we mention the unconventional propagation of quantum correlations [28–30], the distinct entanglement transitions generated by time evolution [31–35] or their extraordinary sensitivity to boundary conditions, also known as the skin effect [36–41]. The latter manifests itself through the unusual localisation of all single-particle eigenstates at the system's edges under open boundary conditions [42–46]. Furthermore, it has unique signatures in the dynamics, leading to non-reciprocal transport [47].

Besides these theoretical developments, unlike their Hermitian counterparts, the toolbox of computational many-body physics is more limited when it comes to non-unitary dynamics. In this work, we introduce a new method to tackle the quantum dynamics of a non-unitary system. Our approach is based on expanding the evolution operator in Faber polynomials [48,49]. This numerical approach is a natural generalisation of its Hermitian counterpart, the Chebyshev polynomial method for time evolution [50], which has been the primary choice for efficiently simulating nonequilibrium transport phenomena in both interacting [51] and non-interacting systems [52–54]. Although time evolution integrators based on Faber polynomials have already been proposed in some works [55–57], for example, in the simulation of electromagnetic wave propagation through passive media [58,59], it appears that their full potential remains largely unexplored. Namely, in the simulation of non-Hermitian quantum systems, they could have a significant impact owing to their numerical stability and adjustable accuracy compared to other methods such as integrators based on Runge-Kutta [60] or Trotterization techniques [61]. Furthermore, the use of Faber Polynomials could replace current techniques [62–64] that still employ Chebyshev polynomials but need to rely on hermitization. This results in the need to operate within a vector space that has twice the original dimensionality.

In this manuscript, we test, benchmark and apply the Faber polynomial method to investigate the time evolution of particle density, charge current, and entanglement in several setups involving the interacting and non-interacting Hatano-Nelson model [65,66], a paradigmatic non-Hermitian model showing non-reciprocity [67] and the skin effect at the single particle level, and non-Hermitian quantum spin chains. In addition, we merge the Faber polynomial method with quantum jumps in order to simulate the dynamics of the full Lindblad master equation through a suitable unravelling and the conditional dynamics encoded in the entanglement of quantum trajectories.

The manuscript is structured as follows. In Sec. 2 we set the stage and define the classes of non-unitary dynamics that we will focus on throughout this manuscript. In Sec. 3 we describe the Faber polynomial method for non-unitary dynamics and discuss its convergence. In Sec. 4 we present our first application to quadratic (Gaussian) non-Hermitian systems. In particular, we explore the melting of a domain-wall state under the Hatano-Nelson Hamiltonian. Furthermore, in Sec. 5, we focus on the spin version of the many-body interacting Hatano-Nelson chain, examining the evolution of both an initial Néel state and a domain wall under the influence of non-reciprocal hopping and interactions. Finally, in Sec. 6, we apply the method to the stochastic quantum jump dynamics obtained by the unravelling of a Lindblad master equation. Sec. 7 summarises our conclusions and discusses potential future research directions.

## 2   Non-unitary dynamics of open quantum systems

In this work, we focus on the dynamics of open quantum many-body systems described by a Hamiltonian $\mathcal{H}$ and a set of independent environments. A typical example will be a quantum spin chain connected on each site to an external bath. In practice, we will always assume a Markovian description of the environment, which can also be identified as a measurement apparatus that monitors a certain physical property of the system, for example the particle density [18,20–22]. However, our primary focus is on the non-unitary dynamics of the system, obtained by tracing out the environment. This is naturally modelled by the Lindblad master equation and its unravelling [10]. In this setting, we consider two types of quantum dynamics: (i) the stochastic evolution of the system conditioned to a given set of measurement outcomes, and (ii) the dynamics of the averaged state. In the first case, the system evolves according to

the stochastic Schrödinger equation,

$$d\,|\psi(\xi_t, t)\rangle = -i\,dt\left[\mathcal{H} - \frac{i}{2}\sum_{\mu}\left(L_{\mu}^{\dagger}L_{\mu} - \langle L_{\mu}^{\dagger}L_{\mu}\rangle_t\right)\right]|\psi(\xi_t, t)\rangle$$

$$+ \sum_{\mu}\left(\frac{L_{\mu}}{\sqrt{\langle L_{\mu}^{\dagger}L_{\mu}\rangle}} - 1\right)d\xi_{\mu,t}\,|\psi(\xi_t, t)\rangle\,, \tag{1}$$

where $\langle \circ \rangle_t \equiv \langle \psi(\xi_t, t)| \circ |\psi(\xi_t, t)\rangle$, and $\xi_t = \{\xi_{\mu,t}\}$ are a set of statistically independent Poisson processes $d\xi_{\mu,t} \in \{0, 1\}$ with average value $\overline{d\xi_{\mu,t}} = dt\langle L_{\mu}^{\dagger}L_{\mu}\rangle_t$. The above dynamics breaks down into two steps: a deterministic non-unitary evolution driven by a non-Hermitian Hamiltonian

$$\mathcal{H}_{\mathrm{nH}} = \mathcal{H} - \frac{i}{2}\sum_{\mu}L_{\mu}^{\dagger}L_{\mu}\,, \tag{2}$$

and a series of stochastic quantum jumps at random times, at which the wave function changes discontinuously (see second line of Eq. (1)). We note that the non-Hermitian evolution is normalised and state dependent. This is encoded in the last term in the first line of Eq. (1). If one post-selects the quantum trajectories over the records of no-click, the dynamics is deterministic and driven by $\mathcal{H}_{\mathrm{nH}}$ [32–34]. Otherwise, if one considers all the trajectories and averages over the measurement outcomes, the conditional density matrix $\rho_c(\xi_t, t) = |\psi(\xi_t, t)\rangle\langle\psi(\xi_t, t)|$, i.e.

$$\rho(t) = \overline{\rho_c(\xi_t, t)}\,, \tag{3}$$

evolves according to the Lindblad master equation with jump operators $L_{\mu}$, i.e.

$$d\rho(t) = -i\,dt\,[\mathcal{H}, \rho] + dt\sum_{\mu}\left(L_{\mu}\rho(t)L_{\mu}^{\dagger} - \frac{1}{2}\left\{L_{\mu}^{\dagger}L_{\mu}, \rho\right\}\right)\,. \tag{4}$$

In both cases, the basic building block of the non-unitary dynamics is the evolution driven by a non-Hermitian Hamiltonian. In the next section, we introduce the Faber polynomial method to accurately solve the time evolution governed by a non-unitary Schrödinger equation.

## 3 Faber polynomial method

The knowledge of the time evolution operator, $\mathcal{U}(t)$, allows a comprehensive description of the physical properties of a system when it is far from equilibrium. This operator is necessary to propagate a given initial state, $|\Psi(t)\rangle = \mathcal{U}(t)|\Psi_0\rangle$, allowing the calculation of observables that characterise the nonequilibrium state. In principle, an exact expression for the state is necessary as soon as one moves beyond the scope of linear response theory. Nevertheless, this problem is equivalent to solving for the spectrum and eigenstates of the Hamiltonian, as in the case of a time-independent Hamiltonian,

$$\mathcal{U}(t) = \exp(-i\mathcal{H}t)\,. \tag{5}$$

The idea behind both the Chebyshev (unitary evolution) and Faber (non-unitary) polynomial methods is to perform an expansion of the time evolution in the respective polynomial basis,

$$\mathcal{U}(t) = \sum_{n=0}^{+\infty}c_n(t)\mathcal{P}_n(\mathcal{H})\,, \tag{6}$$

where $c_n(t)$ is the $n^{\text{th}}$ coefficient of the series expansion and $\mathcal{P}_n$ is the $n^{\text{th}}$ polynomial, which corresponds to a Chebyshev polynomial of the first kind or to a Faber polynomial, depending on the situation. Then the state after the time step, $\delta t$, can be approximated by truncating the series expansion to the order $N_p$,

$$|\Psi(t_0 + \delta t)\rangle \simeq \sum_{n=0}^{N_p-1} c_n(\delta t)|\Psi_n\rangle, \qquad (7)$$

Hamiltonian where we define $|\Psi_n\rangle = \mathcal{P}_n(\mathcal{H})|\Psi(t_0)\rangle$. As will be demonstrated, the coefficients $c_n(\delta t)$ decrease as the order $n$ increases. Moreover, the states $|\Psi_n\rangle$ are efficiently computed through the recurrence relations that the polynomials satisfy. To compute the subsequent level of the expansion, the main computational task is the application of the system's Hamiltonian onto a particular state. Consequently, the most demanding operation involves only multiplying the Hamiltonian by a limited group of vectors, leading to a resource usage that increases linearly with $\dim(\mathcal{H})$ for sparse matrices or quadratically with $\dim(\mathcal{H})$ for dense matrices. Linear scaling is expected for Hamiltonians characterising a system with short-range interactions or hoppings. These principles are exactly those underpinning Kernel Polynomial Methods [68], which has become an essential computational resource in condensed matter physics, particularly for calculating various spectral quantities [69–73]. There has also been established a Non-Hermitian Kernel Polynomial Method [62–64], designed for computing dynamic correlators or spectral functions of the Liouvillian superoperator (or of an effective non-Hermitian Hamiltonian), which still uses Chebyshev polynomials through hermitization techniques.

## 3.1 Warm-up: Unitary evolution

When the time evolution is generated by a Hermitian Hamiltonian, one can expand Eq. (5) using Chebyshev polynomials of the first kind (for further details check Tal-Ezer and Kosloff [50]),

$$\mathcal{U}(t) = \sum_{n=0}^{\infty} c_n(t) T_n(\tilde{\mathcal{H}}), \qquad c_n(t) = \frac{2}{1+\delta_{n,0}}(-i)^n J_n(\lambda t), \qquad (8)$$

where $\tilde{\mathcal{H}} = \mathcal{H}/\lambda$ is the rescaled Hamiltonian, $J_n(x)$ is the $n^{\text{th}}$ Bessel Function of the first kind, and $T_n$ is the $n^{\text{th}}$ Chebyshev polynomial of the first kind. It is necessary to rescale the Hamiltonian so that its eigenvalues fall within the domain of the definition of polynomials, the open interval $(-1, 1)$. It is always possible to do this given that the Hamiltonian is always bounded in finite-size lattice models. Using the recursion relation of the Chebyshev polynomials, the states $|\Psi_n\rangle$ in Eq. (7) are computed on the run using,

$$\begin{aligned}
|\Psi_0\rangle &= |\Psi(t_0)\rangle, \\
|\Psi_1\rangle &= \tilde{\mathcal{H}}|\Psi_0\rangle, \\
|\Psi_{n+1}\rangle &= 2\tilde{\mathcal{H}}|\Psi_n\rangle - |\Psi_{n-1}\rangle, \quad n \geq 2,
\end{aligned} \qquad (9)$$

with $|\Psi_n\rangle = T_n(\tilde{\mathcal{H}})|\Psi\rangle$.

This expansion is feasible solely because the Hamiltonian is Hermitian. Therefore, in the context of open quantum systems where a non-Hermitian operator dictates the dynamics, a different set of polynomials is necessary. Under these circumstances, the spectrum is defined in the complex plane and the operator has right and left eigenvectors, which can be distinct.

## 3.2 Non-unitary evolution

For a non-Hermitian Hamiltonian, the propagator Eq. (5) is expanded using Faber polynomials [74, 75] instead. These are a familiar tool in complex analysis, used as a polynomial basis to

represent a complex-valued function within the domain $\mathcal{D}$ in which it is analytical. The Faber polynomials are generated by conformal mapping, which maps the complement of a closed disk of radius $\rho$ to the complement of the region containing all the spectra of the Hamiltonian (the domain $\mathcal{D}$). In our work, we assume $\mathcal{D}$ to be an elliptic region containing all the eigenvalues of $\mathcal{H}$. This fits our purposes, as the conformal mapping associated with this shape generates a class of Faber polynomials with a minimum recurrence relation (see the Appendix A for further details). The expansion of Eq. (5) for a non-Hermitian Hamiltonian with this choice reduces to

$$\mathcal{U}(t) = \sum_{n=0}^{+\infty} c_n(t) F_n(\tilde{\mathcal{H}}), \qquad c_n(t) = e^{-i\lambda t \gamma_0} \left(\frac{-i}{\sqrt{\gamma_1}}\right)^n J_n\left(2\sqrt{\gamma_1}\lambda t\right), \qquad (10)$$

where $F_n$ is the $n^{\text{th}}$ Faber polynomial. The parameter $\lambda$ is used to rescale the Hamiltonian so that the norm of $F_n(z)$ is bounded [58], and it is obtained from the bounds of the real and imaginary part of the spectra. $\gamma_0$ and, $\gamma_1$ are associated with the details of the elliptic contour chosen, $\gamma_0$ is the centre of the ellipse and $\gamma_1 = 1 - b$, where $b$ is the semi major-axis.[1] The ellipse must be constructed to be close as possible to eigenvalues of $\mathcal{H}$, thereby reducing the magnitude of $\lambda$. One can show [59] that this is achieved using

$$\lambda = \frac{\left(\ell^{2/3} + p^{2/3}\right)^{3/2}}{2}, \qquad \gamma_1 = \frac{\left(\tilde{p}^{2/3} + \tilde{\ell}^{2/3}\right)\left(\tilde{p}^{4/3} - \tilde{\ell}^{4/3}\right)}{4\lambda}, \qquad (11)$$

with $\ell = [\max[\text{Im}(E)] - \min[\text{Im}(E)]]/2$, $p = [\max[\text{Re}(E)] - \min[\text{Re}(E)]]/2$, $\tilde{\ell} = \ell/\lambda$ and $\tilde{p} = p/\lambda$. Using the recursion relation of the Faber Polynomials (consult Appendix A) the states $|\Psi_n\rangle = F_n(\tilde{\mathcal{H}})|\Psi\rangle$ are given by,

$$\begin{aligned}
|\Psi_0\rangle &= |\Psi(t_0)\rangle, \\
|\Psi_1\rangle &= (\tilde{\mathcal{H}} - \gamma_0)|\Psi_0\rangle, \\
|\Psi_2\rangle &= (\tilde{\mathcal{H}} - \gamma_0)|\Psi_1\rangle - 2\gamma_1|\Psi_0\rangle, \\
|\Psi_{n+1}\rangle &= (\tilde{\mathcal{H}} - \gamma_0)|\Psi_n\rangle - \gamma_1|\Psi_{n-1}\rangle, \quad n > 2.
\end{aligned} \qquad (12)$$

In order to perform one time step of evolution, one simply has to truncate Eq. (10) up to the desired order $N_p$, and calculate the associated states $|\Psi_n\rangle$. Through the relations in Eq. (12), one never has to store the Hamiltonian matrix, only needing to store in memory at most the previous two states, $|\Psi_{n-1}\rangle$ and $|\Psi_n\rangle$, to compute the following term of the expansion $|\Psi_{n+1}\rangle$. Furthermore, the expansion coefficients can be computed once at the beginning of the algorithm, as they depend only on the chosen time step. Given this, the computation of the state of the system after a time step scales linearly in the number of polynomials and linearly in the Hilbert space dimension (assuming that the Hamiltonian has a sparse representation on the used basis). This scaling can be improved by using parallelisation techniques [72] and making use of the underlying symmetries of the Hamiltonian [76].

An additional procedure is required when addressing purely non-Hermitian dynamics, specifically, ensuring the normalisation of the quantum state throughout the time evolution. Theoretically, this normalisation can only be done prior to the computation of an observable. However, the algorithm may exhibit instability if the coefficients fluctuate within the limits of machine precision. Consequently, it is prudent to normalise the state following each time step,

$$|\Psi(t + \delta t)\rangle = \frac{\mathcal{U}(\delta t)|\Psi(t)\rangle}{\|\mathcal{U}(\delta t)|\Psi(t)\rangle\|}. \qquad (13)$$

In the following section, we revise how to do this when dealing with fermionic Gaussian states.

---

[1]Recall that the equation of an ellipse is: $\frac{(x-x_0)^2}{a^2} + \frac{(y-y_0)^2}{b^2} = 1$.

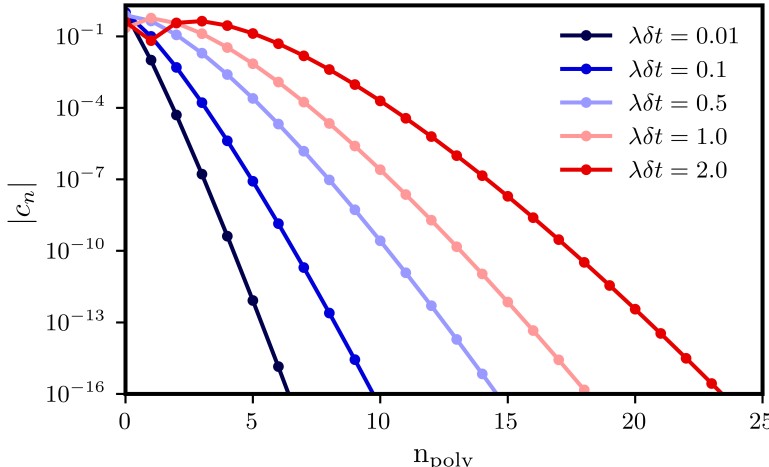

Figure 1: Absolute value of the coefficients associated with the Faber expansion of the time-evolution operator. We represent the absolute of the $n^{th}$ Faber polynomial for different rescaled time steps.

### 3.2.1 Convergence

In this section, we discuss the convergence properties of our algorithm, illustrating in Fig. 1 how the absolute value associated with the $n^{th}$ Faber polynomial varies for different rescaled time steps, $\lambda\delta t$ noting that a greater time step naturally requires more polynomials to achieve convergence. For numerical purposes, the Faber series is an exact representation of the time-evolution operator if the coefficient of the last polynomial used is within the machine precision. This argument holds because of Hamiltonian rescaling, as demonstrated in [77], which guarantees that $\forall_m \max_{z\in G} |F_m(z)| \leq 2$. Using the asymptotic properties of the Bessel functions, we see that the weight of the coefficient decreases with $n$ according to [78],

$$|c_n| \sim \frac{(\lambda\delta t)^n}{n!}, \tag{14}$$

whenever $\lambda\delta t \ll n$. In the remaining of this work, the number of polynomials is chosen such that the last coefficient of the absolute value of the last coefficient of the expansion is of the order $|c_n| \sim 10^{-16}$.

## 4 Application to non-Hermitian Gaussian systems

In this Section, we use the Faber polynomial method to study the dynamics of a fermionic Gaussian non-Hermitian system. Here, the many-body wave function can be expressed using a single-particle basis, and the Faber polynomials can be employed to represent the single-particle propagator. In the case of Hamiltonians with $U(1)$ symmetry, associated with particle number conservation, for an initial Gaussian state with a well-defined particle number M, the many-body state can always be represented in the form

$$|\Psi(t)\rangle = \prod_{n=0}^{M-1}\left[\sum_{\ell=0}^{L-1} U_{\ell n}(t) c_\ell^\dagger\right]|\text{vac}\rangle, \tag{15}$$

with M the total number of particles and $L$ the total number of sites. The time evolution is given by the following equation

$$i\frac{d}{dt}U_n = \mathbf{h}\,U_n, \tag{16}$$

where $\mathbf{U}_n$ is the $n^{th}$ column vector and $\mathbf{h}$ is the single-particle Hamiltonian, a $L \times L$ matrix. This translates the evolution of the many-body state into the evolution of the M single-particle states, each represented by a column in the matrix $\mathbf{U}$. This is expected, as the dynamics and characteristics of a non-interacting system can be simplified to those of the single-particle Hamiltonian, assuming the appropriate quantum statistics. For a typical tight-binding Hamiltonian with a finite number of hopping terms, the complexity of our algorithm for a single time step of evolution scales as $\mathcal{O}(N_p \cdot L \cdot M)$.

Following the time step, it is essential to restore proper normalisation and particle statistics. In the case of Gaussian states, this is achieved through a QR decomposition [18], which guarantees that the U matrix is an isometry, specifically $U^\dagger U = \mathcal{I}_{M \times M}$,

$$U(t + \delta t) = QR, \tag{17}$$

with Q a unitary L×M matrix. So the proper-normalised many-body state is obtained by assigning $U(t + \delta t) = Q$. Although this document focuses on particle-number conserving models, the method can also be easily applied to non-particle conserving Hamiltonians using similar techniques [21, 79]. Typically, this normalisation step is the most computationally intensive part of the time-evolution process. However, unlike other methods, this step can be executed less frequently since the time-step does not need to be small. Thus, this computationally intensive procedure can be minimised while still ensuring high accuracy in the time integration.

The method of directly evolving the state presents a more cost-effective alternative to approaches that solve the equations of motion for all two-point functions. The latter typically employs conventional ordinary differential equation solvers, such as the fourth-order Runge-Kutta method. Firstly, Runge-Kutta methods are only accurate up to a given order. As such, to ensure an accurate time integration, it is necessary to select a short time-step in comparison to the energy scales of the problem, a requirement not imposed by the Faber algorithm. Secondly, the Runge-Kutta method involves the evolution of the full correlation matrix, which in practice involves the evolution of $L \cdot (L-1)/2$ elements. In contrast, the evolution of U requires the evolution of $L \times M$ elements with $M \leq L$. Finally, the Faber algorithm only needs to store two vectors of length L in memory to evolve a given column, while the integration of the correlation matrix demands the storage of four vectors of size L to evolve each of the L columns.

The Faber algorithm is also more beneficial compared to methods that use the exponential of the Hamiltonian. Firstly, one would need to store the exponentiated matrix for repeated application without recalculations. In our approach, it is only necessary to store the system state, since the Hamiltonian matrix is never kept in memory. Additionally, the exponentiated matrix is typically a dense matrix, leading to a computational cost that scales quadratically with its size when multiplied by a vector. Conversely, the Faber Polynomial technique takes advantage of the possible sparcities of the Hamiltonian matrix, making the computational cost linear with respect to the dimension of the single-particle Hilbert space.

## 4.1 Benchmark: Dynamics of the Hatano-Nelson model

The Hatano-Nelson (HN) model [65, 66] is a paradigmatic lattice model for non-Hermitian phenomena. It corresponds to a chain of spinless fermions with an asymmetric hopping,

$$\mathcal{H}_{HN} = -\frac{1}{2} \sum_{\ell=0}^{L-1} \left( (J + \gamma) c_\ell^\dagger c_{\ell+1} + (J - \gamma) c_{\ell+1}^\dagger c_\ell \right), \tag{18}$$

where $J$ is a hopping term to the first-neighbour, $\gamma \in \mathbb{R}$ parameterised the left-right imbalance in charge hopping, also called non-reciprocity, and $c_\ell^\dagger$ ($c_\ell$) is the creation (annihilation) operator which creates (destroys) a fermion on site $\ell$. Under open boundary conditions (OBC),

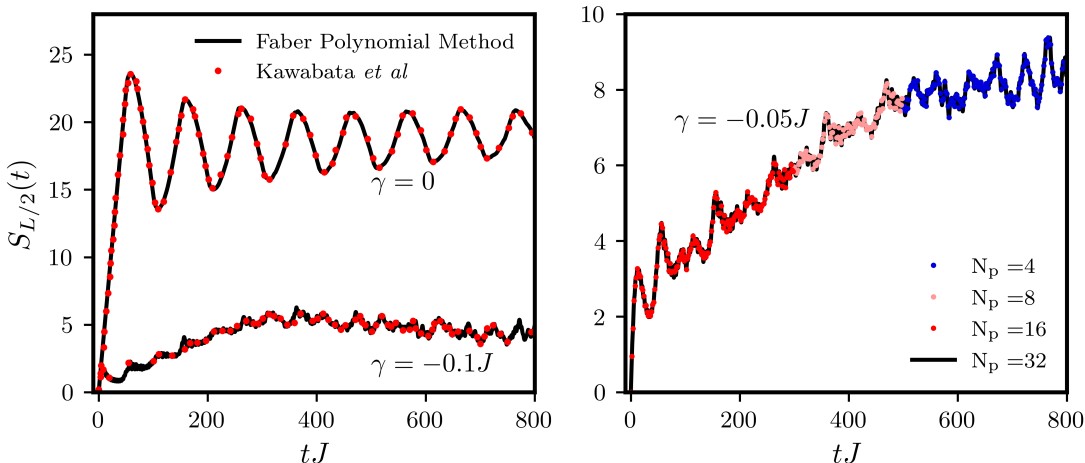

Figure 2: Left panel: Comparison between the results obtained with the Faber polynomial method and those reported in [35] for the entanglement entropy of half of the chain for a total chain of size $L = 100$. Right panel: Entanglement entropy for half of the chain, with $\gamma = -0.05J$ using a different number of polynomials and the time step of $0.1J^{-1}$. We note that we use a symmetric definition for $\gamma$ with regard to [35].

this non-reciprocal hopping gives rise to a unique phenomenon of non-Hermitian systems, the skin effect [36–41]. This corresponds to the localisation of the single-particle eigenstates at the edges of the system. In addition, the model has a huge sensitivity to the boundary conditions: under periodic boundary conditions (PBC), the single-particle spectrum encircles $E = 0$ in the complex plane, with the eigenstates manifesting as delocalised plane waves. In contrast, with OBC, the spectrum is real for $|\gamma| < J$ and purely imaginary for $|\gamma| > J$. Additionally, all right single-particle eigenstates show an exponential localisation at the left (right) boundary for positive (negative) $\gamma$ (check Appendix B for further details).

In the following, we benchmark our results using the Faber polynomial method, with those reported by Kawabata *et al* [35]. That is, we investigate the dynamics of the entanglement [80] associated with a segment of the chain, denoted $\ell$. This is rigorously derived from the von Neumann entropy of the reduced density matrix($\rho_\ell$) [81],

$$S_\ell(t) = -\text{Tr}(\rho_\ell \ln \rho_\ell). \tag{19}$$

$\rho_\ell$ is determined by tracing out the complementary degrees of freedom of the subregion $\ell$. However, the Gaussianity of the state allows us to use the standard techniques [82–84] to perform this computation using the one-particle density matrix restricted to the lattice sites belonging to the region $\ell$, $\mathcal{C}_{n,m\in\ell} = \langle c_n^\dagger c_m \rangle$. Thus, Eq. (19) for a free fermionic system is simplified to

$$S_\ell(t) = -\text{Tr}(\mathcal{C}|_\ell \ln \mathcal{C}|_\ell + (\mathcal{I}_{\ell\times\ell} - \mathcal{C}|_\ell) \ln (\mathcal{I}_{\ell\times\ell} - \mathcal{C}|_\ell)), \tag{20}$$

where $\mathcal{I}_{\ell\times\ell}$ is the $\ell \times \ell$ identity matrix. Similarly to the authors of Ref. [35], we prepare our system in a charge density wave state in the system with open boundaries,

$$|\Psi_0\rangle = \left(\prod_{l=1}^{L/2} c_{2l}^\dagger\right) |\text{vac}\rangle. \tag{21}$$

In Fig. 2 (left) we benchmark the dynamics of the half-chain entanglement entropy $S_{L/2}(t)$ with the results of Ref. [35], for two values of $\gamma$ finding perfect agreement. In the right panel, we

demonstrate convergence with respect to the number of polynomials $N_p$, for a given time-step $\delta t = 0.1 J^{-1}$. We validate the decreasing of the entanglement due to the presence of non-Hermitian Skin effect [35]. In addition to the entanglement entropy, we have also validated our results against other metrics presented in Ref. [35]. For instance, we equally observe the initial charge density wave state rapidly evolving into a state with charge accumulation at one boundary due to non-reciprocal hopping. In Fig. 3 (top panels) we plot the space-time dynamics of particle density as well as a cut at long-times, describing the steady-state density profile along the chain for different system sizes. We see that for short systems, as compared to the single particle wave function localisation length, the accumulation takes the form of a domain wall, while upon increasing system size a finite slope emerges, which we have checked to vanish exponentially with $L$. In Fig. 3 (bottom panels) we plot the dynamics of the local current [35] defined as

$$I_\ell = \frac{Ji}{2} \left\langle c_{\ell+1}^\dagger c_\ell - c_\ell^\dagger c_{\ell+1} \right\rangle. \tag{22}$$

We see that, consistently with the density plot, a finite (negative) current flows in the bulk of the chain, for sufficiently long systems and long times, while at the boundaries the current vanishes due to the localised charge in the domain walls. This current is a feature of this non-equilibrium steady-state, given that it is not present in the ground-state of the Hatano-Nelson model. Besides, in contrast to the Hermitian scenario, the density profiles exhibit a spatial gradient within the bulk of the chain. This non-trivial spatial distribution of density arises from the single-particle skin effect. Additionally, in contrast to the Hermitian scenario, non-Hermiticity permits spatial variations in the local charge current while the density profile remains time-independent. This phenomenon is facilitated by the fact that the continuity equation associated to charge conservation in non-Hermitian systems takes the form

$$\partial_t \left\langle c_\ell^\dagger c_\ell \right\rangle + (I_\ell - I_{\ell-1}) = \mathcal{T}_\ell, \tag{23}$$

where the additional term $\mathcal{T}_\ell$ corresponds to the sink/source of particles due to coupling with the environment equal to,

$$\mathcal{T}_\ell = -\frac{\gamma}{J} \sum_{n=0}^{L-2} \left( \left\langle \left\{ c_\ell^\dagger c_\ell, I_n \right\} \right\rangle - 2 \left\langle I_n \right\rangle \left\langle c_\ell^\dagger c_\ell \right\rangle \right). \tag{24}$$

This term is unique to systems evolving under non-unitary dynamics generated by a non-Hermitian Hamiltonian and has important consequences for the transport properties of these systems [35, 85], as we will further discuss below.

## 4.2 Domain wall melting for Hatano-Nelson

In this Section, our focus is on exploring the impact of non-Hermiticity on the temporal evolution of particle density, current, and entanglement profiles in an HN system initialised in a domain-wall state, $|\text{DW}\rangle = |111\cdots10\cdots000\rangle$.[2] This configuration has been the subject of extensive investigations in the unitary case [88–91], as it exemplifies the distinctive characteristics of non-equilibrium dynamics. In addition, it has inspired the development of generalised hydrodynamics (GHD) [92–94], which facilitates precise calculations of charge and current profiles using a hydrodynamic description. Quantum correlations and entanglement entropy can also be calculated under an extension of this framework, quantum GHD [95]. In the non-Hermitian case, the dynamics of an initial domain wall has been less studied, even in the simple non-interacting HN model.

---

[2]Using the Jordan-Wigner [86, 87] transformation the Hatano-Nelson model can be viewed in the spin-1/2 language as an XX chain with a non-reciprocal XX exchange term.

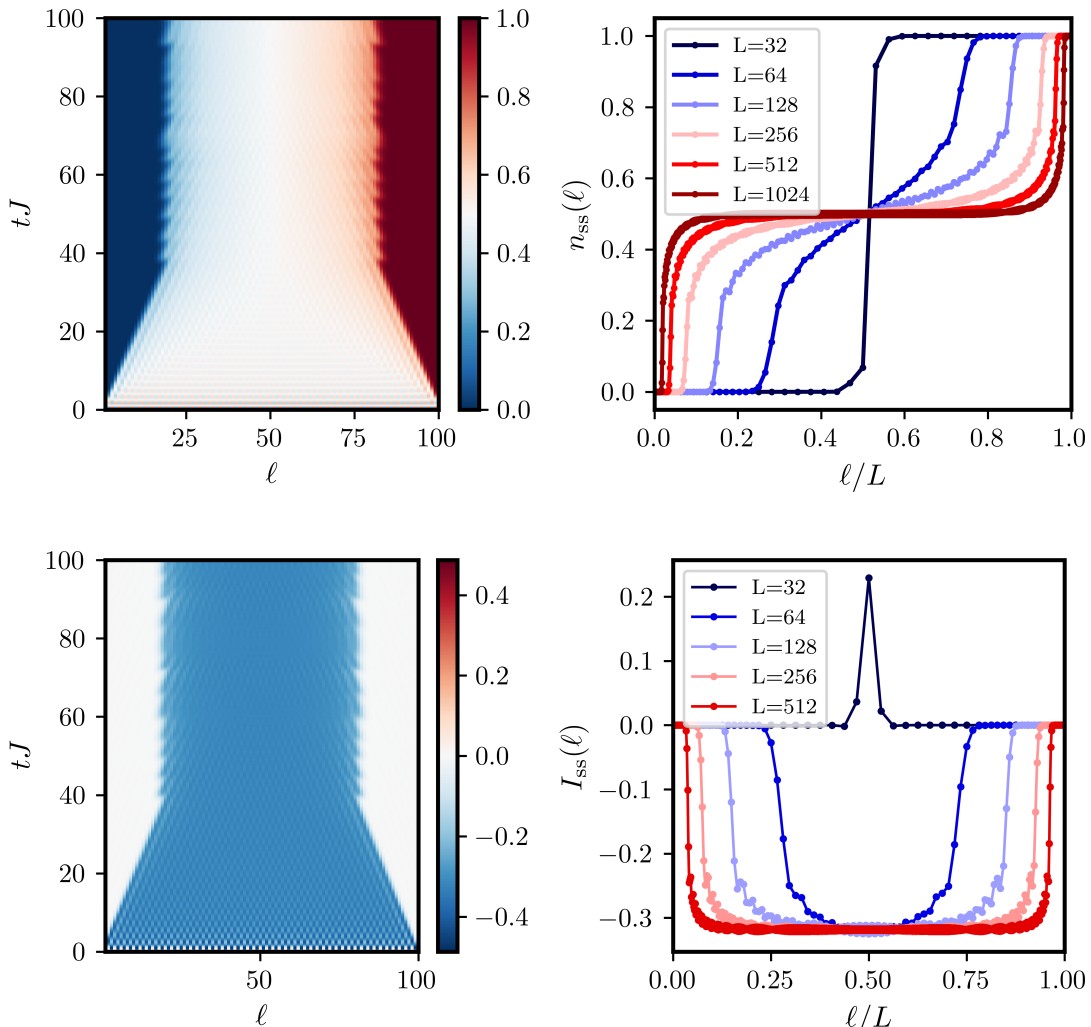

Figure 3: Left - Time and spatial dependence of the particle density (top plot) and charge current (bottom) profile for a system size of 100 sites. Right - Spatial dependence of the particle density (top) and charge current (bottom) in the steady for different system sizes. Other parameters: $\gamma = -0.8J$.

We start considering the time-evolution of the particle density profile under the HN dynamics, that we plot in Fig. 4 for increasing values of the non-reciprocal coupling $\gamma$. In the unitary case ($\gamma = 0$) a clear light cone is visible, corresponding to ballistically propagating quasiparticles. In the non-Hermitian case a light cone is still visible at short times, at least for $|\gamma| < J$, when the particle density satisfies a scaling function $n(\ell/t) = f(\ell/(v_{\text{eff}} t))$, where $v_{\text{eff}}$ is an effective velocity. As the non-reciprocal coupling increases, the light cone shrinks more and more, up to $\gamma = J$, corresponding to the exceptional point of the HN, at which the domain-wall state remains stable. We now focus on the short-time dynamics and discuss the origin of the velocity renormalisation. From the numerical data we obtain (see Fig. 5 (top left))

$$v_{\text{eff}} = J - \gamma. \tag{25}$$

According to this, the domain wall spreads less for higher values of the non-Hermiticity parameter, and the renormalised velocity vanishes at $\gamma = J$. We note that this simple formula is distinct and is not related to those derived for the propagation of wave packets through a

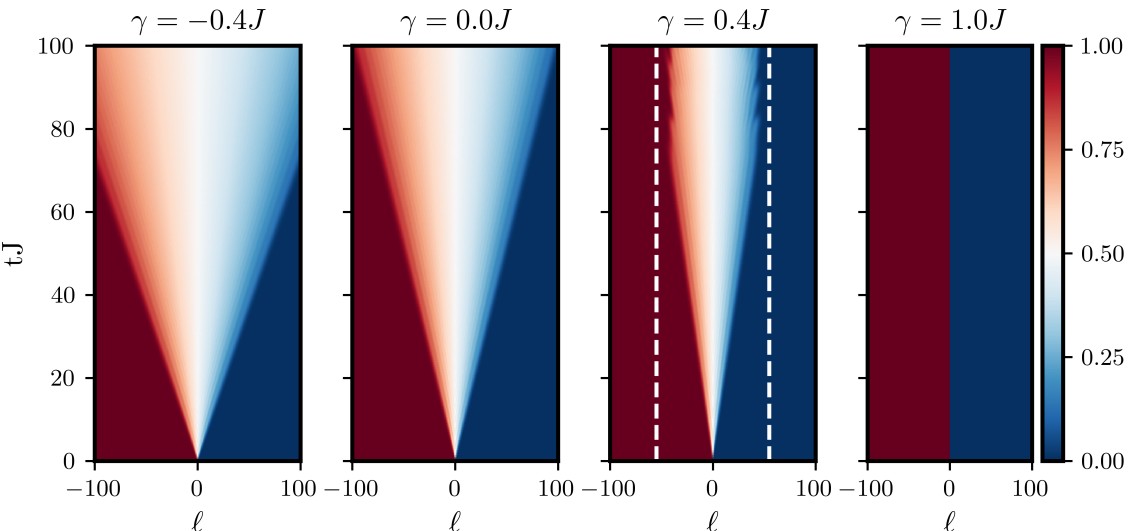

Figure 4: Time evolution of the particle density profile of the Hatano-Nelson model for different values of the non-Hermitian parameter $\gamma$. The total system length corresponds to $L = 256$. The dashed white lines correspond to the effective penetration length.

non-Hermitian medium [96, 97], or to the one obtained from the energy dispersion relation of the model, which would be proportional to $\sqrt{J^2 - \gamma^2}$ (see Fig. 5 (top left) and consult Appendix. B for the explicit derivation). Moreover, it depends on the initial conditions chosen, as if the system was initialised in the state $|\cdots 000111 \cdots\rangle$ instead, the formula should be replaced by $v_{\text{eff}} = J + \gamma$. The reduction in propagation velocity results in a suppression of correlations relative to the Hermitian scenario, a phenomenon previously observed in other non-Hermitian systems [34, 35, 98].

A simple argument to justify the renormalisation of the velocity due to non-hermiticity can be provided. The renormalised velocity can be obtained by using a local continuity equation for the particle density (Eq. (23)) and expanding the term $\mathcal{T}_\ell$. In the non-interacting limit, this term can be expanded using Wick's theorem in a local and non-local term,

$$\mathcal{T}_\ell = -\frac{\gamma}{J}\left(I_\ell + I_{\ell-1}\right)\left(1 - 2\left\langle c_\ell^\dagger c_\ell\right\rangle\right) - i\gamma \sum_{n\neq\{\ell,\ell-1\}}^{L-2} \left[\left\langle c_\ell^\dagger c_{n+1}\right\rangle\left\langle c_n^\dagger c_\ell\right\rangle - \left\langle c_{n+1}^\dagger c_\ell\right\rangle\left\langle c_\ell^\dagger c_n\right\rangle\right]. \quad (26)$$

The second term of this equation is highly non-local and it is zero outside the light-cone region. We proceed by writing the local continuity equation for a site $\ell$, outside the light-cone region, for an instant of time before the arrival of the particle density wavefront. Under these conditions, $I_\ell = 0$ and $\left\langle c_\ell^\dagger c_\ell\right\rangle = 0$ for $\ell > 0$; and $I_{\ell-1} = 0$ and $\left\langle c_\ell^\dagger c_\ell\right\rangle = 1$ for $\ell < 0$. Furthermore, the non-local correlator vanishes, and so, the continuity equation is approximately given by

$$\begin{aligned}
\partial_t \left\langle n_\ell\right\rangle - \left(1 - \frac{\gamma}{J}\right)I_{\ell-1} = 0, \quad \ell > 0, \\
\partial_t \left\langle n_\ell\right\rangle + \left(1 - \frac{\gamma}{J}\right)I_\ell = 0, \quad \ell < 0.
\end{aligned} \quad (27)$$

The $\pm$ sign reflects the different propagation directions. This formula holds only before the particle density wavefront reaches a given site $\ell$. Afterwards, off-diagonal correlations emerge within the light-cone area.

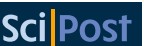

Figure 5: Top - Quasi-particle effective velocity as a function of non-reciprocal coupling (left plot). Particle density profile as a function of space for a fixed time (right plot). Bottom – Spatial profile of charge current at different times for $\gamma = 0.2J$ (right plot). The plot in the bottom left shows the current profile as a function of space and time for $\gamma = 0.8J$. Out of the ballistic region, the renormalised GHD equations are no longer valid. Other parameters: $L = 256$.

Since at short times we can still identify a sharp light cone, it is tempting to try to use a hydrodynamic description for the HN model. Although some advances have been made within the framework of Linblad dynamics [99, 100], formulating a hydrodynamic description for the non-Hermitian variant of the XXZ model remains extremely challenging, even in the absence of interactions. The nonlinearity of the equations of motion results in the loss of most local conservation laws. We proceed in a somewhat phenomenological way and incorporate the renormalised velocity into the hydrodynamic expressions derived for the Hermitian case [101]. Surprisingly, as we show for the Hatano-Nelson model in Fig. 5, the hydrodynamic equations, derived in the unitary problem, still hold in the initial time, as long as the appropriate renormalised velocity is included.

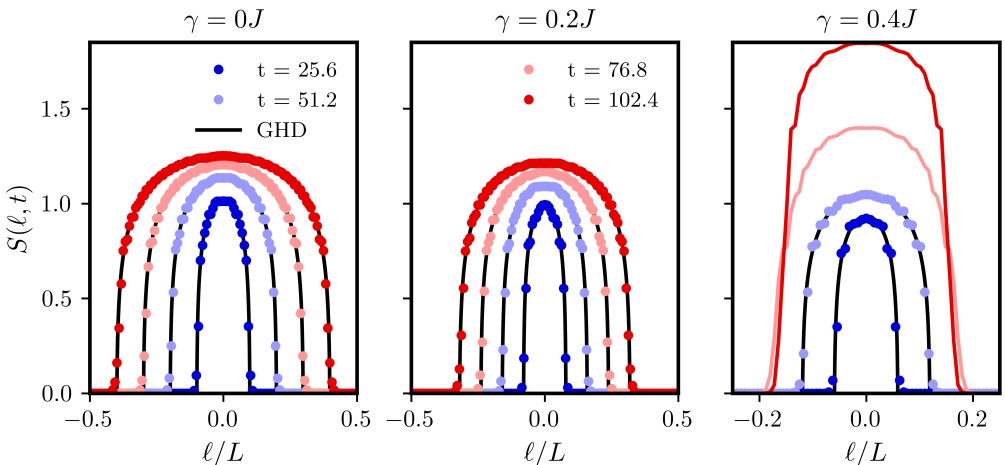

Figure 6: Entanglement entropy for the subsystem $[-L/2, \ell]$ in different times. From right to left, the non-Hermiticity is $\gamma = 0, 0.2J$ and $0.4J$.

With this in mind, the particle density in the spatial interval, $-t \leq \ell \leq t$, is given by the following expression,

$$n(\ell, t) = \frac{1}{\pi} \arccos\left(\frac{\ell}{v_{\text{eff}} t}\right), \tag{28}$$

which perfectly matches the results of the full numerical calculations (see Fig. 5 top right). This is also applicable to the charge current, as we show in Fig. 5 (bottom left). Finally, using the quantum GHD formalism [95, 101], it can likewise be extended to the entanglement entropy of the region $[-L/2, \ell]$,

$$I(\ell, t) = \frac{1}{\pi}\sqrt{1 - \left(\frac{\ell}{v_{\text{eff}} t}\right)^2}, \tag{29}$$

$$S(\ell, t) = \frac{1}{6}\ln\left[v_{\text{eff}} t\left(1 - \left(\frac{\ell}{v_{\text{eff}} t}\right)^2\right)^{3/2}\right] + c_1, \tag{30}$$

where $c_1 \simeq 0.4785$ [102] as shown in the Fig. 6. Similarly to the unitary case, the entanglement increases as a result of the development of quantum correlations within a well-defined spatial region [103]. Ballistic transport of the charge current prevails, leading to a region that expands linearly over time without significant entropy production. This phenomenon is characterised by growth $\ln(t)$, which is emblematic of a local quantum quench protocol [104, 105].

It is important to emphasise that these equations only correctly predict the behaviour at short times, where there is ballistic propagation of the particle density wavefront. At later times, the non-hermiticity plays a greater role than merely renormalising the velocity of propagation.

In the unitary regime, ballistic propagation ceases within temporal scales commensurate with the entire system size, which is applicable to systems of finite dimensions. The non-reciprocal hopping stabilises the domain wall, thus preventing it from melting. With a finite magnitude of non-reciprocal coupling, the particle density wavefront is constrained to penetrate only to a prescribed depth. This maximum penetration depth corresponds to the total system size, $L$, for, $\gamma = 0$ and tends to zero at the exceptional point $\gamma = J$. It is as if the system length is renormalised to an effective one given by

$$L^* \sim \frac{J - \gamma}{J + \gamma} L. \tag{31}$$

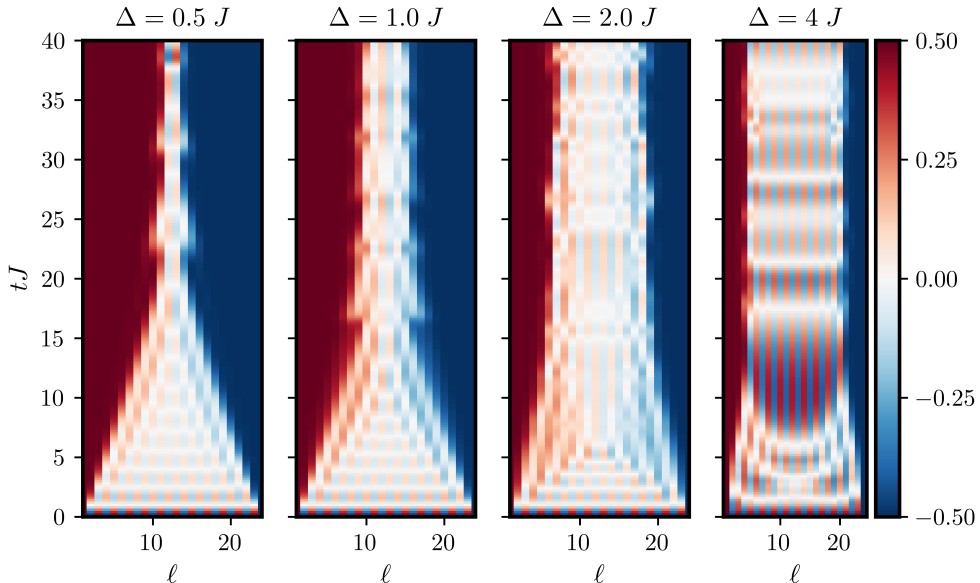

Figure 7: Temporal and spatial dependence of the magnetisation profile, $\langle S_\ell^z \rangle$, for different values of $\Delta$ in the interacting Hatano-Nelson model. Other parameters: $\gamma = 0.8J$ and $L = 24$.

This length is marked by the white dashed lines in Fig. 4. Note that this reasoning is only valid for $\gamma > 0$, since for $\gamma < 0$ the particle density wavefront necessarily reaches the system's boundary. The expression suggests that faster quasiparticles with velocity $v^* = J + \gamma$ cease the ballistic propagation of the particle density wavefront upon hitting the physical boundary. Thus, the maximum propagation velocity allowed in the system increases compared to the Hermitian case, since particles can hop from right to left with velocity $J + \gamma$ (assuming $\gamma > 0$). Following the initial propagation of the particle density wavefront, the system reaches a steady state characterised by the presence of a charge current traversing the two domains. This current emerges due to a flux of particles sourced from the environment, which are subsequently annihilated, as depicted in Fig. 5. However, contrary to the setting described by Kawabata [35], non-Hermiticity leads to the spatial suppression of this current.

## 5 Application to non-Hermitian many-body systems

We now consider an application of the Faber polynomial method to a full non-Hermitian many-body problem. In this case, one can simplify the evaluation of the evolution operator, while the cost of storing the state is still exponential in system size since, as we stress again, in the Faber polynomial method there is no approximation on the state which is fully represented in the basis.

### 5.1 Magnetisation dynamics in the interacting Hatano-Nelson model

In this section, we study the effects of interactions in the dynamics of the magnetisation profile and spin current on a non-Hermitian XXZ chain with a non-reciprocal XX exchange term.

$$\mathcal{H} = -\sum_{\ell=0}^{L-2}\left[ \frac{(J+\gamma)}{2}S_\ell^+ S_{\ell+1}^- + \frac{(J-\gamma)}{2}S_{\ell+1}^+ S_\ell^- + \Delta S_\ell^z S_{\ell+1}^z \right], \tag{32}$$

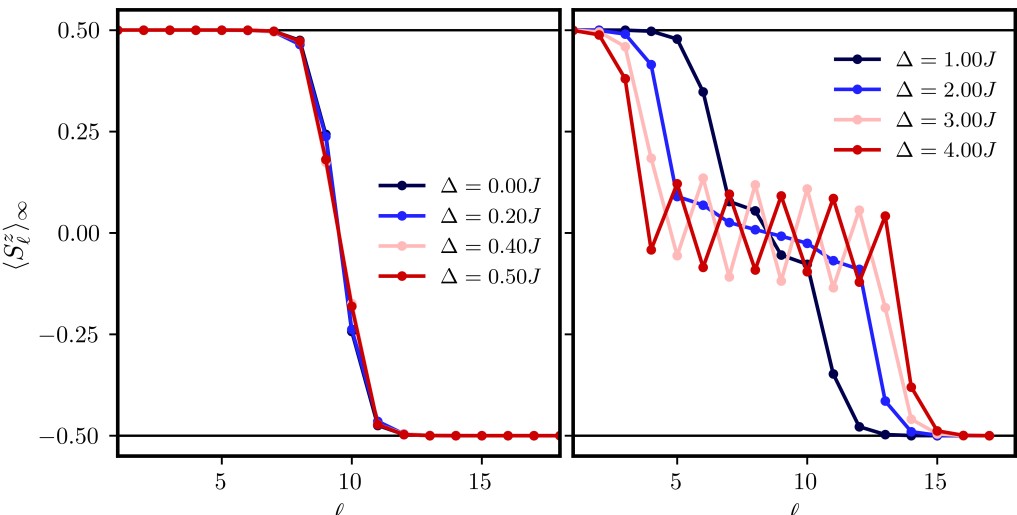

Figure 8: Long-time behaviour of the magnetisation profile shown in Fig. 7, $\langle S_\ell^z \rangle_\infty$, for different values of $\Delta$. We see that at $\Delta = 0$ an emergent domain wall is formed, which is stable for small $\Delta$ (left panel). Increasing $\Delta$, the system develops a potential drop in the middle of the chain, akin to diffusive dynamics, which then further develops into an oscillating patter as $\Delta$ increases (right panel). Other parameters $\gamma = 0.8J$.

where $J$ is a XX exchange term between neighbouring spins, $\gamma$ induces an imbalance between the propagation of left/right magnetic excitations and $\Delta$ is an Ising like exchange. This system can also be viewed as an interacting Hatano-Nelson model by performing the Jordan-Wigner transformation [86, 87]. The system is prepared at time zero in an unentangled Néel state,

$$|\Psi(t = 0)\rangle = |\uparrow, \downarrow, \cdots, \downarrow, \uparrow\rangle, \tag{33}$$

which is an eigenstate of the Ising part of the model, so, one can expect that when $\Delta \gg J, \gamma$ the Néel order is preserved. For $\Delta = 0$, when the model reduces to the non-reciprocal XX chain (or Hatano-Nelson in fermionic language), it is known that the initial Néel state gives rise to a domain wall state at long times as all magnetic excitations are transported to one of the edges of the system [35], just as in the Hatano-Nelson model which exhibits charge accumulation at a boundary. The extent of this proximity is governed by the degree of non-Hermiticity, which is regulated by the parameter $\gamma$. Here, we are interested in understanding the role of interactions in the magnetisation dynamics and the stability of this emergent domain wall. In Fig. 7 we plot the spatial-temporal dynamics of the magnetisation $\langle S_\ell^z(t) \rangle$ for an increasing value of $\Delta$. We see that at short times there is a rapid reshuffling of magnetic excitations driven by the non-reciprocity, towards a boundary accumulation. Interactions compete with this process and tend to preserve the initial antiferromagnetic pattern at the expense of boundary accumulation, as we see well in the right panel of Fig. 7. To better characterise the long-time dynamics, we compute the average magnetisation profile

$$\langle S_\ell^z \rangle_\infty = \lim_{T \to \infty} \frac{1}{T} \int_{\tau^*}^{\tau^* + T} dt \, \langle S_\ell^z(t) \rangle, \tag{34}$$

where $\tau^*$ is the initial time corresponding to the reshuffling of magnetisation until there is a stable boundary accumulation. We plot in Fig. 8 $\langle S_\ell^z \rangle_\infty$ for different values of $\Delta$. We see that the emergent domain-wall state generated by the non-reciprocal exchange is stable at

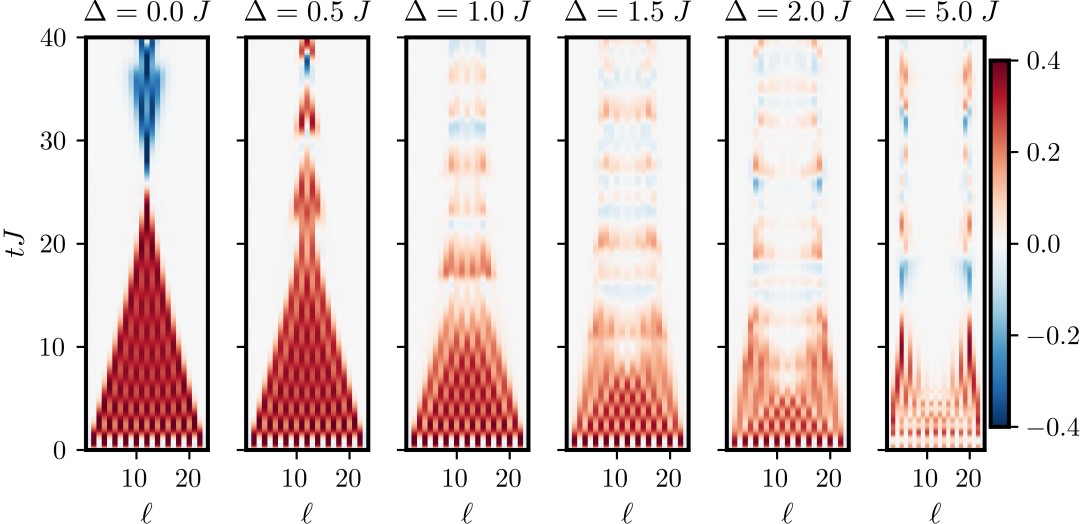

Figure 9: Temporal and spatial dependence of the local current profile, $\langle I_n \rangle$. Other parameters: $\gamma = 0.8J$ and $L = 24$.

weak interactions (left panel). However, as $\Delta$ increases, a novel region emerges that interpolates between the two magnetic domains (right panel). In particular, the system develops a *magnetisation drop*, similar to a potential drop in systems that show diffusive transport. Further increasing $\Delta$, antiferromagnetic correlations emerge on top of this magnetisation slope, which, as expected, are frozen by the large interaction and do not decay away. The result above is particularly intriguing as it suggests that non-reciprocal coupling and interaction collaborate to establish a current-carrying steady state in the system's centre: the former driving the formation of a domain wall that acts as source and drain, the latter providing the necessary scattering term to dissipate and establish a finite average current. Comparable findings have been reported [106] for the ground state of the interacting Hatano-Nelson model with nearest-neighbour repulsion. In that study, they also noted that the initial magnetisation profile (referred to as the real-space Fermi surface) is disrupted by interactions that induce a Néel order. In this work, we demonstrate that such a phenomenon can also be dynamically generated by the non-unitary time-evolution.

Similarly to the non-interacting case, there is a current in this interpolating region, which satisfies the same continuity equation as in Eq. (27). As clearly seen in Fig. 9, once again there is a competition between non-reciprocity and the interaction parameter $\Delta$ in defining the size of this intermediate region, where it is possible for the particle to enter from the environment and give rise to this current. As one is looking at small system sizes, it is not possible to create a stable steady current (like in the non-interacting case). Due to this, we see oscillations in the direction of the central current, and thus there is current flowing in both directions. However, in the extreme case where $\gamma = J$, this oscillatory behaviour ceases to exist, and the current has a fixed direction. When $\Delta$ is much greater than $J$, the current disappears in the bulk region, where the Néel order is maintained.

## 5.2 Effect of interaction in the domain-wall melting for Hatano-Nelson

Finally, we conclude this section by discussing the effect of interactions on the non-Hermitian problem discussed in Sec. 4.2, namely an initial domain wall state.

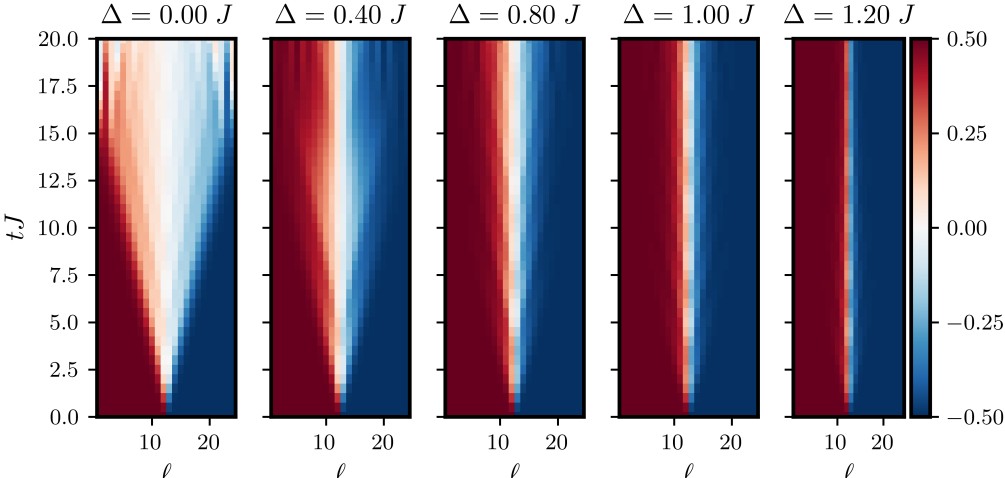

Figure 10: Spatial and temporal dependence of the magnetisation profile for different values of the Ising exchange parameter. Other parameters: spins $\gamma = 0.2J$ and $L = 24$.

In the previous section, we have focused on the domain wall melting for the non-interacting non-reciprocal Hatano-Nelson model, or in the spin analogue, the XX chain as $\Delta = 0$. For a conventional Hermitian XXZ spin chain, the domain wall is only stabilised when the Ising exchange is greater than the XX exchange, $|\Delta| > J$. In contrast, $|\Delta| < J$, the domain wall melts, with a ballistic propagation of the magnetisation wavefront [107]. The Heisenberg point, $\Delta = J$, is special given the existence of the extra spin SU(2) symmetry, which allows for superdiffusive behaviour of the spin current [90, 108]. Nevertheless, there is no Heisenberg point in the spin version of the Hatano-Nelson model, as the non-Hermiticity explicitly breaks the spin SU(2) symmetry. We observe that the interactions contribute to prevent the domain wall from melting, as shown in Fig. 10. In a certain sense, non-Hermiticity and interactions help to preserve the initial magnetic order, which otherwise would be eroded by the dynamics. We have benchmarked this result with matrix product of states (MPS) calculations presented in the Appendix C.

## 6 Application to quantum jumps unravelling

In this Section, we combine the Faber polynomial technique with a high-order Monte Carlo Wave Function algorithm [10, 12] to investigate the Quantum Jumps unravelling of the Lindblad master equation, discussed in Sec. 2. In particular, we address the stochastic Schrödinger equation by propagating the initial state with the Faber polynomial method up to the time instance ($\tau$) when a quantum jump occurs. Therefore, within the time interval $t \in [t_0, t_0 + \tau]$, the state evolves purely non-unitary according to

$$|\psi(t)\rangle = \frac{e^{-i\mathcal{H}t} |\psi(t_0)\rangle}{\|e^{-i\mathcal{H}t} |\psi(t_0)\rangle\|} . \tag{35}$$

The time instance $\tau$ is obtained via the standard higher-order Monte Carlo wave function technique. It corresponds to the specific time at which the norm of the state equates to a random variable, $r$, drawn from a uniform distribution over the interval $[0, 1]$.

SciPost

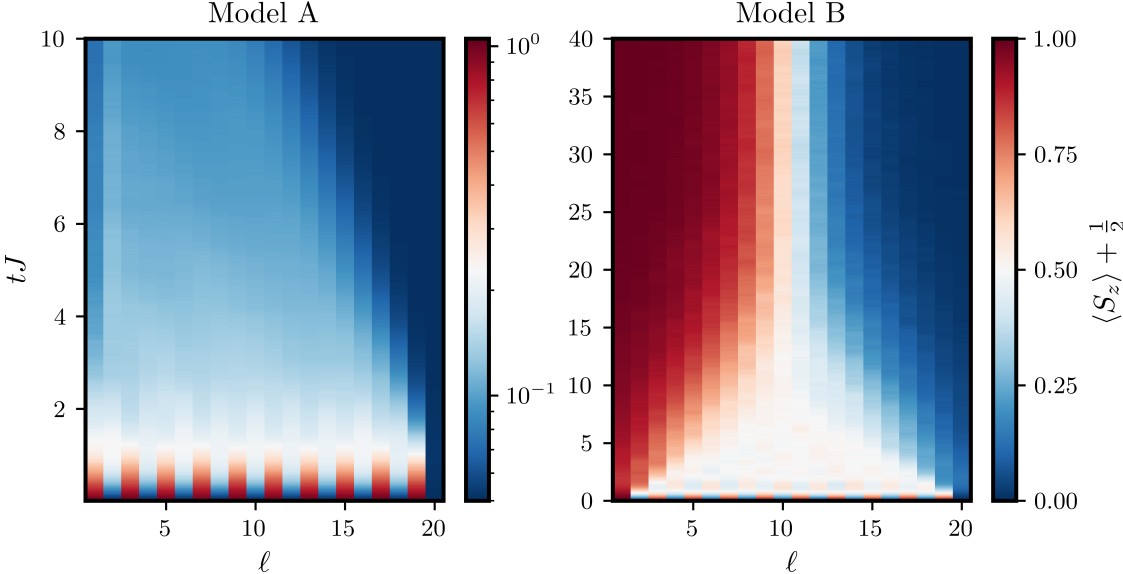

Figure 11: Magnetisation profile in function of the lattice site and time. Left - Dynamics in Model A. Right - Dynamics in Model B. The system was initially prepared in a Néel State (Eq. (33)). Other parameters: $\Delta = 0$, $\gamma = 0.8J$ and $L = 20$.

Consequently, $\tau$ is implicitly defined by the following equation,

$$r = 1 - \langle\psi(t_0)| e^{i\mathcal{H}^\dagger(\tau-t_0)} e^{-i\mathcal{H}(\tau-t_0)} |\psi(t_0)\rangle . \tag{36}$$

The quantum jump is applied by first selecting the quantum jump channel $\mu$ in accordance to their probability

$$p_\mu = \frac{\left\langle L_\mu^\dagger L_\mu \right\rangle}{\sum_\mu \left\langle L_\mu^\dagger L_\mu \right\rangle} , \tag{37}$$

where the average is taken with the state, $|\psi(t+\tau)\rangle$. Then, the post-jump state, $|\psi(t+\tau^+)\rangle$, is obtained by applying the chosen jump operator, $L_\alpha$,

$$|\psi(t+\tau^+)\rangle = \frac{L_\alpha |\psi(t+\tau)\rangle}{\sqrt{\langle\psi(t+\tau)| L_\alpha^\dagger L_\alpha |\psi(t+\tau)\rangle}} . \tag{38}$$

This algorithm gives access to the full Lindbladian dynamics, by averaging over the propagated quantum trajectories. Further, it also provides access to the dynamics under continuous monitoring [10, 21]. This is accomplished by tracking the many-body quantum trajectory and computing, for example, non-linear functions of the state. One such function is the entanglement entropy of quantum trajectories, as we shall discuss below.

## 6.1 Magnetisation and entanglement dynamics in monitored spin chains

As an application, we consider a quantum spin chain described by the Hermitian XXZ model with Hamiltonian

$$\mathcal{H} = -\sum_{\ell=0}^{L-2} \left[ \frac{J}{2} S_\ell^+ S_{\ell+1}^- + \frac{J}{2} S_{\ell+1}^+ S_\ell^- + \Delta S_\ell^z S_{\ell+1}^z \right], \tag{39}$$

where $J$ is the XX exchange term between neighbouring spins and $\Delta$ an Ising like exchange. We compare the dissipative dynamics generated by two different types of quantum jump operators.

A first set of jump operators that we consider describe next-neighbour decoherence of spin excitations along the chain and take the form

$$
\begin{aligned}
L_0 &= \sqrt{|\gamma|}S_0^-\,, \\
L_{1+\ell} &= \sqrt{|\gamma|}\big(S_\ell^- - i\,\mathrm{sgn}(\gamma)S_{\ell+1}^-\big),\, \ell = \{0,\cdots,L-2\}\,, \\
L_L &= \sqrt{|\gamma|}S_{L-1}^-\,.
\end{aligned}
\tag{40}
$$

Through the rest of the document, we name this system Model A. Interestingly, with this choice of jump operators the non-Hermitian Hamiltonian associated to the no-click limit turns out to be given by a spin version of the many-body Hatano-Nelson model. Indeed, the following straightforward calculation yields the non-Hermitian Hamiltonian

$$
\begin{aligned}
\mathcal{H}_{\mathrm{eff}} &= \mathcal{H} - \frac{i}{2}\sum_\ell L_\ell^\dagger L_\ell \\
&= \mathcal{H} - \frac{\gamma}{2}\sum_{\ell=0}^{L-2}\big(S_\ell^+ S_{\ell+1}^- - S_{\ell+1}^+ S_\ell^-\big) - i\,|\gamma|\sum_{\ell=0}^{L-1}S_\ell^+ S_\ell^-\,.
\end{aligned}
\tag{41}
$$

The last term does not affect the dynamics, as it just an overall background decay. Although this model can be connected to the fermionic version of the Hatano-Nelson model, the Linblad equation is not quadratic, hence the quantum trajectories are not gaussian, due to the Jordan-Wigner strings in the quantum jumps, i.e. terms of the form $S^-\rho S^+$.

We compare the dynamics generated by the quantum jumps above with the one induced by a different set of jump operators that create a spin-flip excitation from site $n+1$ to site $n$. These are read as follows,

$$
L_\ell = \sqrt{\gamma}S_\ell^+ S_{\ell+1}^-\,,
\tag{42}
$$

with $\ell = \{0,\cdots,L-2\}$. We refer to this configuration as Model B. Previous studies have shown that these operators induce a phenomenon known as the Liouvillian skin effect [25,109–111]. Specifically, there is an exponential localisation of the Liouvillian modes at the boundaries of the system. This becomes evident when the Linblad is projected onto the one-particle sector, where, at $J = 0$ and $\Delta = 0$, it simplifies to an effective Hatano-Nelson Hamiltonian tuned to its exceptional point [109],

$$
\mathcal{H} = \gamma\sum_{\ell=0}^{L-2}S_\ell^+ S_{\ell+1}^-\,.
\tag{43}
$$

The non-click Hamiltonian differs from the Hatano-Nelson model, representing an XXZ chain with an imaginary Ising exchange term and boundary imaginary magnetic fields,

$$
\mathcal{H}_{\mathrm{eff}} = -\frac{J}{2}\sum_{\ell=0}^{L-2}\big[S_\ell^+ S_{\ell+1}^- + \mathrm{h.c}\big] - \Big[\Delta - i\frac{\gamma}{2}\Big]\sum_{\ell=0}^{L-2}S_\ell^z S_{\ell+1}^z + \frac{i\gamma}{4}\big(S_0^z - S_{L-1}^z\big) - \frac{i\gamma}{8}(L-1)\,.
\tag{44}
$$

In Fig. 11, we plot the magnetisation dynamics starting from an initial Néel state and evolving under the two types of dissipative evolutions. In the left panel, we plot the dynamics generated by Model A (Eq. (40)). Under this set of jump operators, manifestations of non-reciprocity and accumulation of spins in the up state are still present in the transient dynamics [112,113]. We note that these are the same jump operators considered in [113] with the phase, $\phi$, defined by them, tuned to get the non-reciprocal regime. However, in the long time limit, a generic state converges to a configuration with zero excitations, $|\downarrow\cdots\downarrow\rangle$ [113] (see the left plot of Fig. 11). This is attributed to the recycling terms of the form $S_n^-\rho S_n^+$, which effectively remove particles from the system. Additionally, the system has another dark state within the one-magnon sector, which only plays a role in decelerating the relaxation dynamics

towards the fully polarised down state [113]. The final state obtained under this Linbladian does not resemble at all the magnetisation profile that one would have obtained in the no-click limit.

We then focus on Model B (Eq. (42)), and plot in Fig. 11 (right panel) the resulting magnetisation dynamics. For the sake of simplicity, we consider the dynamics with $\Delta = 0$. The dynamics driven by this model will facilitate the accumulation of spins in an up state at the left extremity of the chain. The imaginary Ising exchange term is responsible for diminishing the state's norm when adjacent spins are antialigned, precipitating a quantum jump that propagates a spin towards the left edge. This phenomenon is depicted in the spatial and temporal evolution of the magnetisation profile shown in Fig. 11. In contrast to the behaviour observed in Model A, Model B does not converge to a steady state characterised by zero excitations. This distinction is attributable to a difference in symmetry. Model A exhibits only weak U(1) symmetry, whereas the Model B possesses strong U(1) symmetry, thereby ensuring that the system's state is kept within the initial magnetisation sector at all times.

This disparity in symmetry radically changes the temporal dynamics of the conditionally averaged entanglement entropy, which is defined as the mean entanglement entropy across all conceivable quantum trajectories,

$$\bar{S}_\ell(t) = \int \mathcal{D}\xi_t \, \mathcal{P}(\xi_t) S_\ell(\xi_t) \,, \tag{45}$$

with $S(\xi_t)$ is the entanglement of a given quantum trajectory that evolves according to Eq. (1).

For early times, the entanglement entropy in both models increases linearly. Without non-Hermiticity, the system evolves under standard hermitian dynamics, leading to the entanglement entropy saturating at a value proportional to the system's volume as it locally thermalises [114, 115]. However, in the presence of non-Hermiticity and quantum jumps, this behaviour may drastically alter. In Model A, the entanglement entropy drops to zero after an initial period determined by the measurement rate $\gamma$ [116]. Each jump causes a spin-flip, driving the system to a state devoid of magnetic excitations. This occurs regardless of the system's total size, resulting in a trivial entanglement area law for any nonzero value of $\gamma$. The no-click limit of Model A, which corresponds to the Hatano-Nelson model, also has this area law scaling of the entanglement entropy; however, this is driven by the single-particle skin effect [35].

In contrast, the system in Model B does not relax to a zero excitation state, as seen in Fig. 12. The strong U(1) symmetry confines the dynamics to the magnetisation sector of the initial state.[3] Initially, the system has a linear growth of the entanglement entropy, reminiscent of the unitary evolution. However, similarly to the Hatano-Nelson model, the steady-state supports an area law entanglement for a positive value of $\gamma$, as seen in the left plot of Fig.12. This can be understood through the spectral properties of the no-click Hamiltonian (Eq. (44)). The spectrum is always complex-valued, and so, in the no-click limit, the steady-state corresponds to right-eigenstate with the slowest decaying mode in the zero magnetisation sector. In particular, for certain values of $\gamma/J$, the imaginary component of the spectrum is gapped. Thus, the entanglement entropy inevitably follows an area law in the long-time limit, similar to the ground state of a gapped Hamiltonian [33, 117]. The imaginary gap ($\Delta_{\mathrm{Im}}$) in the zero magnetisation sector can be analytically obtained in the limit of $\gamma \gg J$,

$$\Delta_{\mathrm{Im}} = -i\frac{\gamma}{2J} + \mathcal{O}\left(\frac{J}{\gamma}\right). \tag{46}$$

---

[3]The dynamics conserves the total number of excitations in the initial state as the initial state is eigenstate of $\sum_{n=0}^{L-1} S_n^z$ and the Hamiltonian in Eq. (44) as U(1) symmetry.

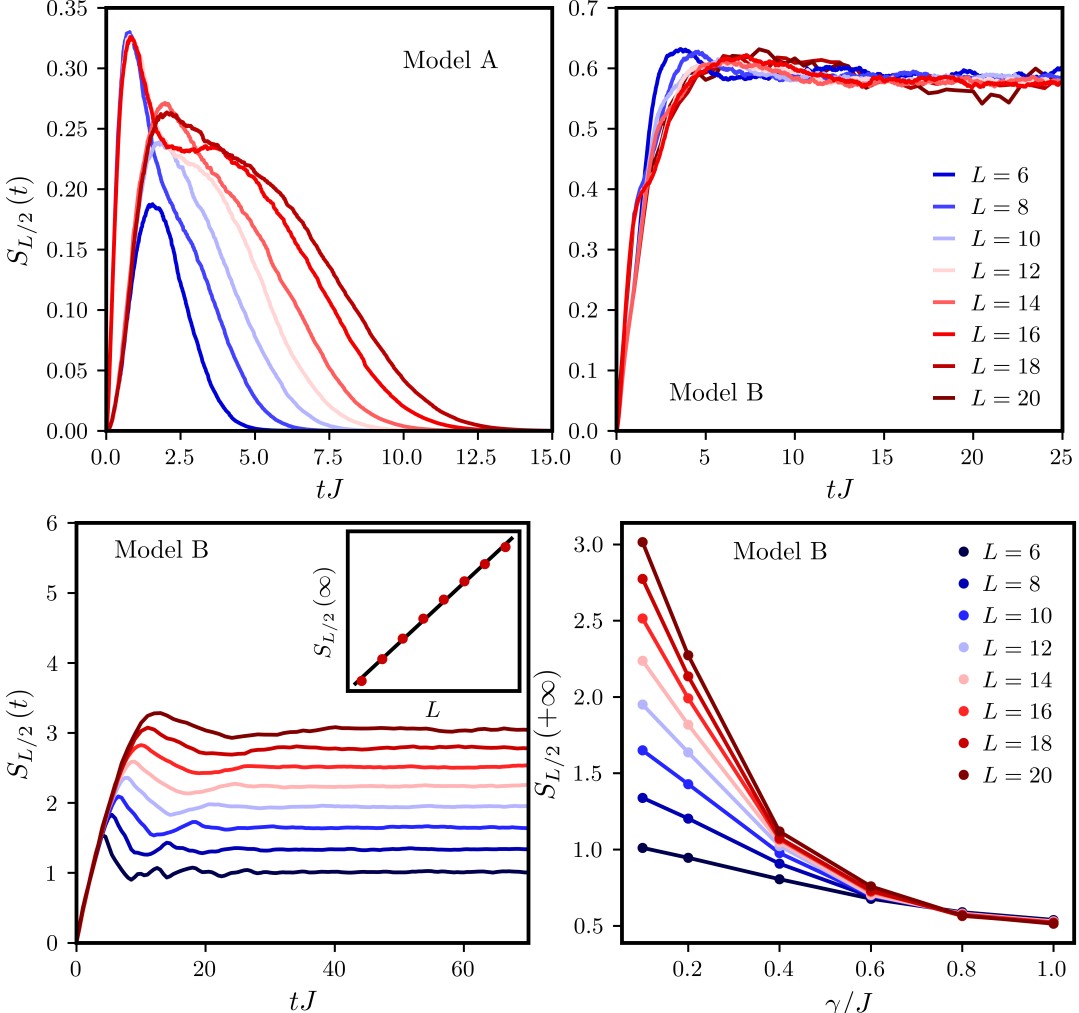

Figure 12: Top - Time dependence of the conditional entanglement entropy for different system sizes with $\gamma = 0.8J$. On the left, the dynamics corresponds to Model A, see Eq. (40)), while on the right to Model B, see Eq. (42). Bottom Left: Time evolution of the entanglement entropy for $\gamma = 0.1J$ in the case of Model B. The inset corresponds to the fit of the steady-state entanglement entropy to the law $S_{L/2}(\infty) = a_0 + a_1 L$, with $a_0 = (0.145 \pm 0.002)$ and $a_1 = (0.18 \pm 0.03)$. Bottom Right - Steady-state entanglement entropy in Model B, for different system sizes as a function of the non-hermitian parameter. Other parameters: $\Delta = 0.0J$.

The measurement apparatus effectively disentangles the system, pushing excitations towards the left boundary, and preventing the formation of long-range correlations. It is harder to completely confirm this for smaller values of $\gamma$, where the unitary evolution dominates both the non-Hermitian and stochastic terms. This mainly affects systems with a smaller dimension, where finite-size effects are substantial, since the terms proportional to $L^{-n}$ with $n > 0$ in the entanglement entropy cannot be ignored. For example, for $\gamma = 0.1J$, one cannot extrapolate the true entanglement scaling, as the observed linear growth might be an artefact of small system sizes, so it remains uncertain whether a genuine entanglement transition occurs in the thermodynamic limit. Nevertheless, the remaining data points clearly indicate the collapse of all system sizes to the same value, thus revealing the area law nature of the entanglement entropy.

# 7 Conclusions

Throughout this study, we have successfully used the Faber polynomial method to characterise the non-unitary dynamics of both non-interacting and interacting Hatano-Nelson models. Additionally, we have seamlessly integrated this approach with a high-order Monte Carlo Wave Function algorithm to rigorously examine both Lindblad and continuous monitoring dynamics.

In the non-interacting problem, we provided the first numerical evidence supporting the existence of a valid hydrodynamic description for the melting dynamics of a domain wall in the presence of non-Hermitian terms. This finding encourages further developments to properly formalise a theory of generalised hydrodynamics applicable to non-Hermitian systems.

Our study also reveals an intriguing competition between the Ising exchange term, which tends to preserve the initial Néel order, and the non-reciprocal XX coupling, which tends to form a domain-wall ordering. For comparable values of $\Delta$ and $J$, the interaction allows the formation of an intermediate region that interpolates between the two magnetic domains, allowing the flow of current. However, we could not reach considerable system sizes to determine if this region could support a non-equilibrium steady-state current flowing in only one direction, as seen in the non-interacting case. It is clear, however, that this cannot be the case for $\Delta \gg J$, as the dynamics preserve the initial magnetic ordering, and the current can only exist in the interpolating region between the Néel-ordered domain and the ferromagnetic one generated by the non-reciprocal coupling. Conversely, our work shows that interactions and non-reciprocity help preserve the initial magnetic order when the system is initialised in a domain-wall setup, a result consistent with both the non-interacting non-Hermitian problem and the interacting Hermitian case.

This study offers additional insights into the entanglement transition in quantum spin chains exhibiting Non-Hermitian or Liouvillian skin effects. In Model A, we found that the area law behaviour of entanglement entropy remains for any nonzero $\gamma$, similar to the no-click limit. However, the sources are different: in the no-click limit, the area law stems from the single-particle skin effect, while in the monitored stochastic trajectory, it is due to the quantum jumps that relax the system into the fully polarised down spin-state, a product state. On the other hand, the dynamics with two-body jump operators (Model B), $L_\ell \propto S_\ell^+ S_{\ell+1}^-$, allow for a non-equilibrium steady state with a magnetisation profile resembling the no-click limit of the Hatano-Nelson model. Furthermore, the average entanglement entropy still follows an area law for finite values of the ratio $\gamma/J$ in conditional dynamics. The measurement apparatuses effectively disentangle the state, suppressing the volume law that would otherwise be generated in unitary dynamics. This work extends previous studies that focused on the one-particle sector [35,109].

Overall, our results support the utility and applicability of Faber polynomials in various research domains. This encompasses investigations into measurement-induced phase transitions, open quantum systems, and the exploration of purely non-unitary dynamics governed by non-Hermitian Hamiltonians. For instance, Faber Polynomials could be employed to study non-Hermitian Floquet problems in the high-frequency limit, where the Floquet Hamiltonian is well-defined and time-independent. Moreover, these polynomials can also be used to calculate general spectral properties of non-Hermitian Hamiltonians, potentially replicating the role of Chebyshev polynomials within the Kernel polynomial method [68]. This would complement the existing Non-Hermitian Kernel polynomial method [62–64]. Furthermore, the Faber polynomial approach could potentially be combined with MPS, similar to the developments already made in the Hermitian case [118–121].

# Acknowledgments

**Funding information**  R.D.S. acknowledges funding from Erasmus+ (Erasmus Mundus programme of the European Union) and from the Institute Quantum-Saclay under the project *QuanTEdu-France*. M.S. acknowledges funding from the European Research Council (ERC) under the European Union's Horizon 2020 research and innovation program (Grant agreement No. 101002955 - CONQUER). We acknowledge Collège de France IPH cluster where the numerical calculations were carried out.

# A   Further details on Faber polynomials

As discussed previously, Faber polynomials serve as a polynomial basis to represent complex-valued functions that are analytic within the domain $\mathcal{D}$. These are generated by a conformal mapping $\xi(w)$ that maps the complement of a closed disk of radius $\rho$ to the complement of $\mathcal{D}$,

$$\frac{\xi'(w)}{\xi(w)-z} = \sum_{n=0}^{\infty} \frac{1}{w^{n+1}} F_n(z)\,, \tag{A.1}$$

with $F_n(z)$ being the $n^{th}$ Faber polynomial generated by the conformal mapping $\xi(w)$, $z \in \mathcal{D}$ and $w$ is such that $|w| > \rho$. The existence of such a map, which also satisfies the conditions $\xi(w)/w \to 1$ in the limit $|w| \to \infty$, is guaranteed by the Riemann mapping theorem [58,122]. Furthermore, $\xi(w)$ admits a Laurent expansion at $w = \infty$ of the form,

$$\xi(w) = w + \sum_{m\geq 0} \gamma_m w^{-m}\,, \tag{A.2}$$

where $\gamma_m \in \mathbb{C}$. Using the Laurent expansion and integrating in the contour defined around the disk of radius $\rho$, it is straightforward to check that the Faber polynomials satisfies the following recurrence relation,

$$F_{n+1}(z) = zF_n(z) - \sum_{j=0}^{n} \gamma_j F_{n-j}(z) - n\gamma_n\,, \quad n > 0\,, \tag{A.3}$$

where $F_0(z) = 1$. For our purposes, we are interested in using the Faber polynomials to approximate a given function of our non-Hermitian Hamiltonian, $f(\mathcal{H})$. The domain $\mathcal{D}$, is defined by the spectrum of the Hamiltonian. Using Eq. (A.1), the expression in a Faber series is given by

$$f(z) := \sum_{k=-\infty}^{+\infty} c_k F_k(z), \tag{A.4}$$

with the coefficients given by the integral,

$$c_n = \frac{1}{2\pi i} \int_{|w|=\rho} \frac{f(\xi(w))}{w^{n+1}} dw\,. \tag{A.5}$$

As stated in the main text, we perform this integral with $\rho = 1$ by properly rescaling the Hamiltonian. Furthermore, we compute the Faber coefficients related to an elliptic contour. For this contour, the conformal mapping reduces to $\xi(w) = w + \gamma_0 + \gamma_1 w^{-1}$, where $\gamma_0$ is the centre of the ellipse and $\gamma_1 = 1 - b$, with $b$ the major semiaxis. This maximises the memory

efficiency of the algorithm, as the recurrence relation in Eq. (A.3) only depends on the two previous polynomials,

$$
\begin{aligned}
F_0(z) &= 1\,, \\
F_1(z) &= z - \gamma_0\,, \\
F_2(z) &= (z - \gamma_0)\,F_1(z) - 2\gamma_1 F_0(z)\,, \\
F_{n+1}(z) &= (z - \gamma_0)\,F_n(z) - \gamma_1 F_{n-1}(z)\,, \quad n \geq 2\,.
\end{aligned}
\tag{A.6}
$$

The coefficients $c_n$ of the Faber series presented in the main text (Eq. (10)) are straightforwardly computed by performing the contour integral in Eq. (A.5) for the function $f(z) = e^{-i\delta t_s z}$. This is done through the use of the identity [78],

$$
\exp\left(\frac{z}{2}\left[s + \frac{a}{s}\right]\right) = \sum_{n=-\infty}^{+\infty} \left(\frac{s}{i\sqrt{a}}\right)^n J_n\left(i\sqrt{a}z\right)\,.
\tag{A.7}
$$

The Faber Polynomials have two interesting limits; when $\gamma_1 = 0$ the conformal mapping corresponds to a circle, thus the Faber Polynomials reduce to the Taylor polynomials centred around $\gamma_0$,

$$
F_n(z) = (z - \gamma_0)^n\,,
\tag{A.8}
$$

Whereas, in the limit $\gamma_1 = 1$, the conformal mapping evolves the real line, and so, the Faber Polynomials can be related with the Chebyshev polynomials by

$$
\begin{aligned}
F_1(z) &= T_1\left(\frac{z - \gamma_0}{2}\right)\,, \\
F_2(z) &= T_2\left(\frac{z - \gamma_0}{2}\right) - 1\,, \\
F_n(z) &= T_n\left(\frac{z - \gamma_0}{2}\right)\,, \quad n \geq 2\,.
\end{aligned}
\tag{A.9}
$$

# B General features on the Hatano-Nelson model

In this Appendix, we review the essential characteristics of the non-interacting Hatano-Nelson model, highlighting the importance of boundary conditions on the resulting physical properties. Specifically, we examine the diagonalisation of the non-interacting Hatano-Nelson model (Eq. (18)) under both open and periodic boundary conditions. For OBCs, the Hamiltonian can be diagonalised through a similarity transformation. This is done using the $GL(1)$ gauge transformation [35],

$$
\hat{p}_\ell^\dagger := e^{\ell\theta}\hat{c}_\ell^\dagger\,, \qquad \hat{q}_\ell := e^{-\ell\theta}\hat{c}_\ell\,,
\tag{B.1}
$$

with $\theta \in \mathbb{R}$ and both $p_\ell$ and $q_\ell$ two fermionic operators satisfying the unusual anti-commutation relations: $\{p_\ell^\dagger, q_n\} = \delta_{\ell,n}$, $\{p_\ell^\dagger, p_n\} = e^{2\ell\theta}\delta_{\ell,n}$, $\{q_\ell^\dagger, q_n\} = e^{-2\ell\theta}\delta_{\ell,n}$ and $\{p_\ell, q_n\} = \{p_\ell^\dagger, q_n^\dagger\} = 0$. This indicates the biorthogonality of the Hamiltonian's eigenbasis.

The Hamiltonian can subsequently be expressed in the form

$$
\mathcal{H}_{\mathrm{HN}} = -\sum_{n=0}^{L-2}\left[\frac{e^\theta\,(J+\gamma)}{2}p_n^\dagger q_{n+1} + \frac{e^{-\theta}\,(J-\gamma)}{2}p_{n+1}^\dagger q_n\right]\,.
\tag{B.2}
$$

The $\theta$ parameter is chosen so that the final Hamiltonian becomes hermitian,

$$
\theta = \frac{1}{2}\log\left(\frac{J-\gamma}{J+\gamma}\right)\,.
\tag{B.3}
$$

This reduces the Hamiltonian to

$$\mathcal{H}_{\mathrm{HN}} = -\frac{\sqrt{J^2 - \gamma^2}}{2} \sum_{n=0}^{L-2} \left[ p_n^\dagger q_{n+1} + p_{n+1}^\dagger q_n \right] . \tag{B.4}$$

The Hamiltonian can be diagonalised through a straightforward Fourier transformation,

$$\mathcal{H}_{\mathrm{HN}} = -\sqrt{J^2 - \gamma^2} \sum_k \cos(k) p_k^\dagger q_k , \tag{B.5}$$

where $k = \frac{\pi}{L+1} n$, $n \in \{1, \cdots, L\}$. The quasi-particles generated by $p_k^\dagger$ and $q_k^\dagger$ are nonorthogonal, as shown by their anticommutation relations. Furthermore, they correspond to states exponentially localised at the left and right edges of the chain,

$$\begin{aligned}
p_k^\dagger |\mathrm{vac}\rangle &= \sqrt{\frac{2}{L+1}} \sum_{\ell=0}^{L-1} e^{\ell\theta} \sin(k \cdot (\ell+1)) c_\ell^\dagger |\mathrm{vac}\rangle , \\
q_k^\dagger |\mathrm{vac}\rangle &= \sqrt{\frac{2}{L+1}} \sum_{\ell=0}^{L-1} e^{-\ell\theta} \sin(k \cdot (\ell+1)) c_\ell^\dagger |\mathrm{vac}\rangle .
\end{aligned} \tag{B.6}$$

The parameter $\theta$ controls wave-function localisation, inducing a characteristic length scale $l$,

$$l \sim \left( \log\left(\frac{J-\gamma}{J+\gamma}\right) \right)^{-1} . \tag{B.7}$$

This characteristic length scale emerges due to the skin effect and is exclusive to the open-boundary condition scenario. Conversely, under PBCs, the Hamiltonian can be diagonalised using a Fourier transform without employing the GL gauge transformation,

$$\mathcal{H}_{\mathrm{HN}} = -\sum_{k \in \mathrm{FbZ}} \left[ J \cos(ka) + i\gamma \sin(ka) \right] c_k^\dagger c_k . \tag{B.8}$$

Unlike the previous case where the eigenstates were confined to the chain ends, under periodic boundary conditions, both the right and left eigenstates are indistinguishable and delocalised. Furthermore, the spectra is always complex for all values of $\gamma$.

## C  Comparison with MPS based methods

In this Appendix, we compare the Faber Polynomial technique with MPS calculations for the interacting domain-wall melting problem. First, using a second-order time-evolution block decimation (TEBD) algorithm [123], and second, a first-order matrix product operator (MPO) representation of the time-evolution operator [124]. These methods were implemented using the ITensor Library [125, 126]. The TEBD algorithm consists in performing a Suzuki-trotter break-up of the time evolution operator,

$$e^{-i\delta t(\mathcal{H}_{\mathrm{even}} + \mathcal{H}_{\mathrm{odd}})} \simeq e^{-i\delta t \mathcal{H}_{\mathrm{even}}} e^{-i\delta t \mathcal{H}_{\mathrm{odd}}} , \tag{C.1}$$

where the Hamiltonian in Eq. (32) is decomposed as a sum of two body terms either acting on the left or right sites, $\mathcal{H} = \mathcal{H}_{\mathrm{odd}} + \mathcal{H}_{\mathrm{even}}$. While in the first-order MPO, we use a Euler expansion of the time-evolution operator, $e^{-i\delta t \mathcal{H}} \simeq 1 - i\delta t \mathcal{H}$, and represent the Hamiltonian through an MPO.

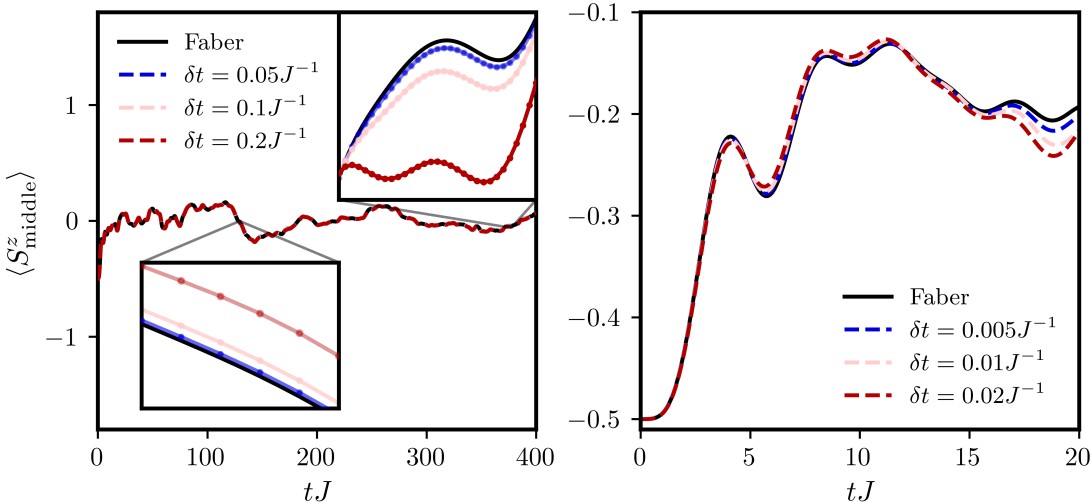

Figure 13: Comparison between the Faber polynomial expansion and the TEBD (MPO) algorithm, shown on the left (right). The calculations on the left were performed with $\gamma = 0.0J$, $\Delta = 0.5J$ and $L = 18$, while on the right we used $\gamma = 0.2J$, $\Delta = 0.4J$ and $L = 18$.

MPS calculations could be effective for this problem, as the entanglement entropy increases logarithmically over time and the domain-wall order is favoured by non-Hermiticity. However, the TEBD and MPO algorithms face errors that the Faber algorithm avoids. Firstly, there is a restriction on the maximum allowed time step. In the TEBD algorithm, the Hamiltonian is decomposed into smaller two-body terms that can be exactly exponentiated, as shown in Eq. (C.1). The approximation neglects the non-zero commutator term $[\mathcal{H}_{\text{odd}}, \mathcal{H}_{\text{even}}]$, causing an error of order $\delta t^2$. To minimise this error, the time step must be small relative to the problem's energy scale. This is evident in the left panel of Fig. 13, where longer integration times with a larger time step deviate from results obtained with Faber polynomials. A comparable error is present in the MPO algorithm due to the use of a first-order expansion. The Faber algorithm circumvents this issue by precisely representing the time evolution operator for any time step with any desired accuracy, though it requires the computation of the appropriate number of polynomials. Another limitation of the TEBD algorithm is the handling of long-range interactions with the Suzuki-Trotter decomposition, which requires the use of advanced techniques such as swap gates [127]. The TEBD and MPO techniques also face inaccuracies because of the truncation of the MPS bond dimension, a problem that the Faber algorithm avoids at the cost of working with a state encompassing the entire Hilbert space dimension. Of course, using the Faber polynomial limits the analysis to smaller system sizes. However, it can be advantageous in situations where MPS calculations fail, such as when the entanglement entropy scales proportionally to the system size.

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
