# Peer review of "Non-Unitary Quantum Many-Body Dynamics using the Faber Polynomial Method"

_SciPost Physics, doi:SciPost Phys. 17, 128 (2024)_

## Round 2 · Referee Report · Anonymous (Referee 1) · 2024-8-19

Report
The authors introduce the use of Faber polynomials to study the time evolution of non-Hermitian systems. This is analogous to the use of Chebyshev polynomials in the Hermitian case. They then apply this method to a number of examples including non-interacting and interacting models as well as to study full quantum trajectories in an unraveling of the Lindblad equation. The paper is well written and introduces a useful method for non-Hermitian quantum systems. The examples are of interest and the obtained results are discussed in detail. Once the points below are addressed, the paper seems to be suitable for SciPost Physics.
I have the following comments:
1) It might be worth to mention already in the introduction that one possibility is also to use Hermitization and then use standard Hermitian methods such as Chebyshev polynomials. Of course, this also includes a doubling of the vector space which is a disadvantage but it should nevertheless be mentioned.
2) In the non-interacting case where the Hamiltonian is not time dependent: Can one not just simply exponentiate and use exact diagonalization? Ist there any advantage in using Faber polynomials in this case? Related: What method is used in Ref. [34] which is used for a comparison in Fig. 2? Is it ED?
3) It is not clear to me why GHD would be applicable here. GHD is used for integrable one-dimensional quantum systems and relies on the fact that such models have an infinite set of local conservation laws. The model here seems to be non-integrable and such local conserved charges do not exist. So why should a generalized GHD description be possible? Related: Is Eq. (30) just a phenomenological fit formula or is there an underlying theory to motivate this form of the entanglement?
4) Fig. 12 and discussion of this Fig: I find it difficult to see which panel shows which of the two models and have the same issue with the discussion. For example, the sentence "This occurs regardless ... area law for any nonzero value of \gamma." Which of the two models does this refer to?
5) Fig. 12: If I understand correctly, then the lower two panels in Fig. 12 show the model with the S^+S^- dissipator. Are the authors arguing that the entanglement entropy always shows a volume law for all \gamma? Why can it be excluded that there is a transition from volume to area law entanglement at some critical \gamma_c?
6) App. B: Is the solution with k=pi/(L+1) *n correct for both L even and L odd? I do not quite recall the solution for this model but remember that often one gets a transcendental equation in one case and the above simple solution in the other.
7) Fig. 13: Here the system size is L=18 but the methods are described as TEBD and MPO. Since the system size is so small, I assume that no truncation is necessary. If this is the case: Would it not be more precise to call the former a Trotter-Suzuki method and the latter an exact MPO representation?
There are some small typos throughout the paper. I just list, as examples, the following:
i) Below Eq. (17): Even tough, thought ...
ii) Below Eq. (22): non-equilibrium steady, ...
iii) Above Eq. (26): by working the term T_\ell.
Requested changes
See above.
Recommendation
Ask for minor revision
We thank the Referee for their time in reviewing our manuscript, as well as for the raised questions.
The Referee pointed out some questions and requests to be addressed before publication. Here we comment on their questions and requests (following their numbering), pointing out how we accordingly improved our manuscript.
1) It might be worth to mention already in the introduction that one possibility is also to use Hermitization and then use standard Hermitian methods such as Chebyshev polynomials. Of course, this also includes a doubling of the vector space which is a disadvantage but it should nevertheless be mentioned.
- We have included a comment about hermitization techniques in the introduction and another comment in Sec. 3.0, when we mention the Kernel Polynomial Method.
2) In the non-interacting case where the Hamiltonian is not time-dependent: Can one not just simply exponentiate and use exact diagonalization? Is there any advantage in using Faber polynomials in this case? Related: What method is used in Ref. [34] which is used for a comparison in Fig. 2? Is it ED?
- The computational effort of using the Faber Polynomials is lower in computational and memory resources. Using ED and simply exponentiating would have the following drawbacks: i) The process requires performing exact diagonalization on a non-Hermitian matrix, which has a computational complexity that scales with the cube of the single-particle Hilbert space dimension. Additionally, this approach is prone to significant numerical instabilities, especially when the system is near an exceptional point. ii) It is necessary to store the exponentiated matrix so that it can be applied without needing to recalculate it. In our implementation, we only need to store the state of the system, as the Hamiltonian matrix is never stored in memory. iii) The exponentiated matrix is, in principle, a dense matrix, so multiplying it by a vector scales with the square of its dimension. However, in the Faber Polynomial method, we take advantage of the sparsity of the Hamiltonian matrix, making the computational effort linear in the dimension of the single-particle Hilbert space. In Ref. 34, they are using ED, we have added included this point in our previous discussion at the end of sec.4.0.
3) It is not clear to me why GHD would be applicable here. GHD is used for integrable one-dimensional quantum systems and relies on the fact that such models have an infinite set of local conservation laws. The model here seems to be non-integrable and such local conserved charges do not exist. So why should a generalized GHD description be possible? Related: Is Eq. (30) just a phenomenological fit formula or is there an underlying theory to motivate this form of the entanglement?
- We thank the Referee for raising this point. We agree that, a priori, the applicability of GHD to a non-Hermitian model is surprising. We have modified the discussion in Section 4.2 to clarify this point more effectively. As we now emphasize more clearly in the text, GHD is used only as a phenomenological description of our results, and we leave the question of a full theoretical derivation for future work. However, we stress that the rationale behind this result is that GHD applies only at short times, where, as shown in Figure 4, the dynamics display a clear light-cone. This suggests that in the short-time dynamics, the non-Hermiticity of the Hamiltonian does not play a major role, aside from renormalizing the light-cone velocity (as discussed in Eq. 26). Consequently, one could use all the known results for the melting of a domain wall in the unitary case for the dynamics of magnetization, entanglement, and current, by including the properly renormalized velocity. In our approach (Eqs. 28–30), we use the unitary expressions with the appropriately renormalized velocity. Our numerical results show that this is remarkably accurate at short times
4) Fig. 12 and discussion of this Fig: I find it difficult to see which panel shows which of the two models and have the same issue with the discussion. For example, the sentence "This occurs regardless ... area law for any nonzero value of \gamma." Which of the two models does this refer to?
- We have named the models Model A and Model B to make it clear in the discussion which model we are referring to.
5) Fig. 12: If I understand correctly, then the lower two panels in Fig. 12 show the model with the S^+S^- dissipator. Are the authors arguing that the entanglement entropy always shows a volume law for all \gamma? Why can it be excluded that there is a transition from volume to area law entanglement at some critical \gamma_c?
- From the numerical data obtained, we observe that in the weak monitoring phase, with gamma = 0.1J, the system sizes used exhibit volume scaling. For all other values of gamma, finite-size analysis indicates that the entanglement entropy follows area-law scaling for L >> 1. In the main text, we mention that, based on the accessible system sizes, it is difficult to determine whether the observed volume scaling is stable and robust. This could be a consequence of the small system sizes studied, and with further increases in total system size, one might observe that this phase, in the thermodynamic limit, actually corresponds to an area law.
6) App. B: Is the solution with k=pi/(L+1) *n correct for both L even and L odd? I do not quite recall the solution for this model but remember that often one gets a transcendental equation in one case and the above simple solution in the other.
- We have confirmed that the solution is the same for both L even and odd.
7) Fig. 13: Here the system size is L=18 but the methods are described as TEBD and MPO. Since the system size is so small, I assume that no truncation is necessary. If this is the case: Would it not be more precise to call the former a Trotter-Suzuki method and the latter an exact MPO representation?
In our numerical simulations, we were truncating but keeping the truncation error bellow 1e-12. However, for this system's size, the main source of error should come from the approximation scheme used.

Author: Rafael Diogo Soares on 2024-09-13 [id 4779]
(in reply to Report 2 on 2024-08-27)We thank the Referee for their time spent reviewing our manuscript. The Referee pointed out some questions. Here we comment on those, pointing out how we accordingly improved our manuscript:
1) The authors discuss expectation values calculated within the right basis, which is fine. Is it possible to generalize this method to the biorthogonal basis as well?
2) In addition to quenches, would it be possible to study time dependent ramps or even time periodic protocols with a sufficient modification of the method?

---

## Round 3 · Referee Report · Anonymous (Referee 1) · 2024-10-5

Report

All the points in my previous report have been addressed. In particular, the limits of the applicability of GHD for the non-hermitian domain wall melting problem have been clarified which was my most important concern.

I have no further comments and recommend that the paper is published in its current form.

Recommendation

Publish (easily meets expectations and criteria for this Journal; among top 50%)

---

## Round 3 · List of Changes

• We have included a comment about hermitization techniques in the introduction and another in Sec. 3.0, where we mention the Kernel Polynomial Method.

  • We have added a general comment on the advantages of the method compared to methods that rely on the exponentiation of the Hamiltonian.

  • Following Referee 1's comment, we have more clearly emphasized in the text that GHD is used only as a phenomenological description of our results.

  • We have named the two models used in Sec. 6 and restructured the discussion and figures to clarify which results refer to each model.

  • We have included a comment on the applicability of the method to non-Hermitian Floquet problems in the high-frequency limit, where a time-independent Floquet Hamiltonian can be defined.

---

## Editorial Decision

published